**Neoglacial Climate Anomalies and the Harappan Metamorphosis**
Authors:
Liviu Giosan[1]*, William D. Orsi[2,3], Marco Coolen[4], Cornelia Wuchter[4],
Ann G. Dunlea[1], Kaustubh Thirumalai[5], Samuel E. Munoz[1], Peter D. Clift[6],
Jeffrey P. Donnelly[1], Valier Galy[7], Dorian Q. Fuller[8]
Affiliations:
[1]Geology & Geophysics, Woods Hole Oceanographic Institution, Woods Hole, MA, USA
[2]Department of Earth and Environmental Sciences, Paleontology & Geobiology, Ludwig-
Maximilians-Universität München, Munich, Germany
[3]GeoBio-CenterLMU, Ludwig-Maximilians-Universität München, Munich, Germany
[4]Faculty of Science and Engineering, Curtin University, Perth, Australia
[5]Department of Earth, Environmental, and Planetary Sciences, Brown University, Providence,
RI, USA
[6]Geology & Geophysics, Louisiana State University, USA
[7]Marine Chemistry & Geochemistry, Woods Hole Oceanographic Institution, Woods Hole, MA,
USA
[8]Institute of Archaeology, University College London, London, UK
*Correspondence: lgiosan@whoi.edu

Abstract:
Climate exerted constraints on the growth and decline of past human societies but our knowledge
of temporal and spatial climatic patterns is often too restricted to address causal connections. At
a global scale, the inter-hemispheric thermal balance provides an emergent framework for
understanding regional Holocene climate variability. As the thermal balance adjusted to gradual
changes in the seasonality of insolation, the Inter-Tropical Convergence Zone migrated
southward accompanied by a weakening of the Indian summer monsoon. Superimposed on this
trend, anomalies such as the Little Ice Age point to asymmetric changes in the extratropics of
either hemisphere. Here we present a reconstruction of the Indian winter monsoon in the Arabian
Sea for the last 6000 years based on paleobiological records in sediments from the continental
margin of Pakistan at two levels of ecological complexity: sedimentary ancient DNA reflecting
water column environmental states and planktonic foraminifers sensitive to winter conditions.
We show that strong winter monsoons between ca. 4,500 and 3,000 years ago occurred during a
period characterized by a series of weak interhemispheric temperature contrast intervals, which
we identify as the Early Neoglacial Anomalies (ENA). The strong winter monsoons during ENA
were accompanied by changes in wind and precipitation patterns that are particularly evident
across the eastern Northern Hemisphere and Tropics. This coordinated climate reorganization
may have helped trigger the metamorphosis of the urban Harappan civilization into a rural
society through a push-pull migration from summer flood-deficient river valleys to the
Himalayan piedmont plains with augmented winter rains. The decline in the winter monsoon
between 3300 and 3000 years ago at the end of ENA could have played a role in the demise of
the rural late Harappans during that time as the first Iron Age culture established itself on the
Ghaggar-Hakra interfluve. Finally, we speculate that time-transgressive landcover changes due
to aridification of the Tropics may have led to a generalized instability of the global climate
during ENA at the transition from the warmer Holocene Thermal Maximum to the cooler
Neoglacial.


1.   Introduction
The growth and decline of human societies can be affected by climate (e.g., Butzer, 2012;
DeMenocal, 2001) but addressing causal connections is difficult, especially when no written
records exist. Human agency sometimes confounds such connections by acting to mitigate
climate pressures or, on the contrary, increasing the brittleness of social systems in face of
climate variability (Rosen, 2007). Moreover, our knowledge of temporal and spatial climatic
patterns remains too restricted, especially deeper in time, to fully address social dynamics.
Significant progress in addressing this problem has been made especially for historical intervals
(e.g., Carey, 2012; McMichael, 2012; Brooke, 2014; Izdebski et al., 2015; d'Alpoim Guedes et
al., 2016; Nelson et al., 2016; Ljungqvist, 2017; Haldon et al., 2018) using theoretical
reconsiderations, novel sources of data and sophisticated deep time modeling that could lead to
better consilience between natural scientists, historians and archaeologists. The coalescence of
migration phenomena, profound cultural transformations and/or collapse of prehistorical
societies regardless of geographical and cultural boundaries during certain time periods
characterized by climatic anomalies, events or regime shifts suggests that large scale climate
variability may be involved (e.g., Donges et al., 2015 and references therein). At the global scale,
the interhemispheric thermal balance provides an emergent framework for understanding such
major Holocene climate events (Boos and Korty, 2016; Broecker and Putnam, 2013; McGee et
al., 2014; Schneider et al., 2014). As this balance adjusted over the Holocene to gradual changes
in the seasonality of insolation (Berger and Loutre, 1991), the Inter-Tropical Convergence Zone
(ITCZ) migrated southward (e.g., Arbuszewski et al., 2013; Haug et al., 2001) accompanied by a
weakening of the Indian summer monsoon (e.g., Fleitmann et al., 2003; Ponton et al., 2012).
Superimposed on this trend, centennial- to millennial-scale anomalies point to asymmetric
changes in the extratropics of either hemisphere (Boos and Korty, 2016; Broccoli et al., 2006;
Chiang and Bitz, 2005; Chiang and Friedman, 2012; Schneider et al., 2014).
The most extensive but least understood among the early urban civilizations, the Harappan (Fig.
1 and 2; see supplementary materials for distribution of archaeological sites), collapsed ca. 3900
years ago (e.g., Shaffer, 1992). At their peak, the Harappans spread over the alluvial plain of the
Indus and its tributaries, encroaching onto the Sutlej-Yamuna or Ghaggar-Hakra (G-H) interfluve
that separates the Indus and Ganges drainage basins (Fig. 1). In the late Harappan phase that was
characterized by more regional artefact styles and trading networks, cities and settlements along
the Indus and its tributaries declined while the number of rural sites increased on the upper G-H
interfluve (Gangal et al., 2001; Kenoyer, 1998; Mughal, 1997; Possehl, 2002; Wright, 2010). The
agricultural Harappan economy showed a large degree of versatility by adapting to water
availability (e.g., Fuller, 2011; Giosan et al., 2012; Madella and Fuller, 2006; Petrie et al., 2017;
Weber et al., 2010; Wright et al., 2008). Two precipitation sources, the summer monsoon and
winter westerlies (Fig. 1), provide rainfall to the region (Bookhagen and Burbank, 2010; Petrie et
al., 2017; Wright et al., 2008). Previous simple modeling exercises suggested that winter rain
increased in Punjab over the late Holocene (Wright et al., 2008). During the hydrologic year, part
of this precipitation, stored as snow and ice in surrounding mountain ranges, is redistributed as
meltwater by the Indus and its Himalayan tributaries to the arid and semi-arid landscape of the
alluvial plain (Karim and Veizer, 2002).
The climatic trigger for the urban Harappan collapse was probably the decline of the summer
monsoon (e.g., Dixit et al., 2014; Kathayat et al., 2017; MacDonald, 2011; Singh et al., 1971;
Staubwasser et al 2003; Stein, 1931) that led to less extensive and more erratic floods making
inundation agriculture less sustainable along the Indus and its tributaries (Giosan et al., 2012)
and may have led to bio-socio-economic stress and disruptions (e.g., Meadow, 1991; Schug et
al., 2013). Still, the remarkable longevity of the decentralized rural phase until ca. 3200 years
ago in the face of persistent late Holocene aridity (Dixit et al., 2014; Fleitmann et al., 2003;
Ponton et al., 2012; Prasad and Enzel, 2006) remains puzzling. Whether the Harappan
metamorphosis was simply the result of habitat tracking toward regions where summer monsoon
floods were still reliable or also reflected a significant increase in winter rain remains unknown
(Giosan et al., 2012; Madella and Fuller, 2006; Petrie et al., 2017; Wright et al., 2008). To
address this dilemma, we present a proxy record for the Indian winter monsoon in the Arabian
Sea and show that its variability was an expression of large scale climate reorganization across
the eastern Northern Hemisphere and Tropics affecting precipitation patterns across the
Harappan territory. Aided by an analysis of Harappan archaeological site redistribution, we
speculate that the Harappan relocation after the collapse of its urban phase may have conformed
to a push-pull migration model.
2. Background
Under modern climatological conditions (Fig. 3), the summer monsoon delivers most of the
precipitation to the former Harappan territory, but winter rains are also significant in quantity
along the Himalayan piedmont (i.e., between 15 and 30% annually). Winter rain is brought in
primarily by extra-tropical cyclones embedded in the Westerlies (Dimri et al., 2015) and are
known locally as Western Disturbances (WD). These cyclones distribute winter rains to a zonal
swath extending from the Mediterranean through Mesopotamia, the Iranian Plateau and
Baluchistan, all and across to the western Himalayas (Fig. 3). Stronger and more frequent WD
rains in NW India are associated with southern shifts of the Westerly Jet in the upper troposphere
(e.g., Dimri et al. 2017). Surface winter monsoon winds are generally directed towards the
southwest but they blow preferentially toward the east-southeast along the coast in the
northernmost Arabian Sea (Fig. 3). An enhanced eastward zonal component over the northern
Arabian Sea is typical for more rainy winters (Dimri et al. 2017). Although limited in space and
time, modern climatologies indicate a strong, physical linkage between winter sea-surface
temperatures (SST) in the northern Arabian Sea and precipitation on the Himalayan piedmont,
including the upper G-H interfluve (see also supplementary materials). Ultimately, the thermal
contrast between the cold Asian continent and relatively warmer Indian Ocean is thought to be
the initial driver of the Indian monsoon winds (Dimri et al., 2016).
In contrast to the wet summer monsoon, winds of the winter monsoon flow from the continent
toward the ocean and are generally dry. That explains in part why Holocene reconstructions of
the winter monsoon are few and contradictory, suggesting strong regional variabilities (Jia et al.,
2015; Kotlia et al., 2017; Li and Morrill, 2015; Sagawa et al., 2014; Wang et al., 2012; Yancheva
et al., 2007). Holocene eolian deposits linked to the winter monsoon are also geographically-
limited (Li and Morrill, 2015). However, in the Arabian Sea indirect wind proxies based on
changes in planktonic foraminifer assemblages and other mixing properties have been used to
reconstruct distinct hydrographic states caused by seasonal winds (Böll et al., 2014; Curry et al.,
1992; Lückge et al., 2001; Munz et al., 2015; Schiebel et al., 2004; Schulz et al., 2002). Winter
monsoon winds blowing over the northeast Arabian Sea cool its surface waters via evaporation
and weaken thermal stratification promoting convective mixing (Banse and McClain, 1986; Luis
and Kawamura, 2004). Cooler SSTs and the injection of nutrients into the photic zone lead in
turn to changes in the plankton community (Madhupratap et al., 1996; Luis and Kawamura,
2004; Schulz et al., 2002). To reconstruct the history of winter monsoon we thus employed
complementary proxies for convective winter mixing, at two levels of ecological complexity: (a)
sedimentary ancient DNA to assess the water column plankton community structure, and (b) the
relative abundance of *Globigerina falconensis*, a planktonic foraminifer sensitive to winter
conditions (Munz et al.; 2015; Schulz et al., 2002).
3. Methods
3.1 Sediment Core
We sampled the upper 2.3 m, comprising the Holocene interval, in the 13-m-long piston core
Indus 11C (Clift et al., 2014) retrieved during *R/V Pelagia* cruise 64PE300 in 2009 from the
oxygen minimum zone (OMZ) in the northeastern Arabian Sea (23°07.30'N, 66°29.80'E; 566 m
depth) (Fig. 1). The chronology for the Holocene section of the core was previously reported in
Orsi et al. (2017) and is based on calibrated radiocarbon dates of five multi-specimen samples of
planktonic foram *Orbulina universa* and one mixed planktonic foraminifer sample. Calibration
was performed using Calib 7.1 program (Stuiver et al., 2018) with a reservoir age of $565 \pm 35$
radiocarbon years following regional reservoir reconstructions by Staubwasser et al. (2002).
Calibrated radiocarbon dates were used to derive a polynomial age model (see supplementary
materials). The piston corer did not recover the last few hundred years of the Holocene record
probably due to overpenetration. However, indistinct but continuous laminations downcore with
no visual or X-radiograph discontinuities, together with the radiocarbon chronology indicate that
the sedimentary record recovered is continuous.

## 3.2. Ancient DNA Analyses

A total of five grams of wet weight sediment were extracted inside the ancient DNA-dedicated lab at Woods Hole Oceanographic Institution (WHOI), aseptically as described previously (Coolen et al., 2013) and transferred into 50 mL sterile tubes. The sediments were homogenized for 40 sec at speed 6 using a Fastprep 96 homogenizer (MP Biomedicals, Santa Ana, CA) in the presence of beads and 15 ml of preheated (50 °C) sterile filtered extraction buffer (77 vol% 1M phosphate buffer pH 8, 15 vol% 200 proof ethanol, and 8 vol% of MoBio's lysis buffer solution C1 [MoBio, Carlsbad, CA]). The extraction was repeated with 10 ml of the same extraction buffer but without C1 lysis buffer (Orsi et al., 2017). After centrifugation, the supernatants were pooled and concentrated to a volume of 100 μl without loss of DNA using 50,000 NMWL Amicon® Ultra 15 mL centrifugal filters (Millipore) and contaminants were removed from the concentrated extract using the PowerClean® Pro DNA Clean-up Kit (MoBio). The exact same procedures were performed in triplicate without the addition of sediment as a control for contamination during extraction and purification of the sedimentary DNA.

The extracted and purified sedimentary DNA was quantified fluorometrically using Quant-iT PicoGreen dsDNA Reagent (Invitrogen), and ~20 nanograms of each extract was used as template for PCR amplification of preserved planktonic 18S rRNA genes. The short (~130 base pair) 18S rDNA-V9 region was amplified using the domain-specific primer combination 1380F (5'-CCC TGC CHT TTG TAC ACA C-3') and 1510R (5'CCT TCY GCA GGT TCA CCT AC-3')(Amaral-Zettler et al., 2009). Quantitative PCR was performed using a SYBR®Green I nucleic acid stain (Invitrogen) and using a Realplex quantitative PCR system (Eppendorf, Hauppauge, NY). The annealing temperature was set to 66 °C and all reactions were stopped in the exponential phase after 35-42 cycles. 18S rRNA libraries were sequenced on an Illumina MiSeq sequencing using the facilities of the W.M. Keck Center for Comparative and Functional Genomics, University of Illinois at Urbana-Champaign, IL, USA sequenced 18S libraries that resulted in approximately 12 million DNA sequences.

The 18S rRNA gene sequences were processed using the Quantitative Insights Into Microbial Ecology (QIIME) environment (Caporaso et al., 2010). Reads passing quality control (removal of any sequence containing an 'N', minimum read length 250 bp, minimum Phred score=20) were organized into operational taxonomic units (OTUs) sharing 95% sequence identity with UCLUST (Edgar et al., 2010) and assigned to taxonomic groups through BLASTn searches against the SILVA database (Pruesse et al., 2007). OTU tables were rarefied to the sample with the least number of sequences, and all OTUs containing less than one sequence were removed. OTUs that were detected in only one sample were also removed. Metagenomes were directly sequenced bi-directionally on an Illumina HiSeq, at the University of Delaware Sequencing and Genotyping Center (Delaware Biotechnology Institute). Contigs were assembled de novo as described in Orsi et al. (2017). To identify contigs containing chlorophyll biosynthesis proteins,

open reading frames on the contig sequences were detected using FragGeneScan (Rho et al.,
2010), and protein homologs were identified through BLASTp searches against the SEED
database (www.theseed.org).  Only hits to reference proteins with at least 60% amino acid
similarity over an alignment length >50 amino acids were considered true homologs and used for
downstream analysis. Assignment of ORFs to biochemical pathway classes were made based on
the SEED metabolic pathway database and classification scheme. The relative abundance of
reads mapping to ORFs was normalized against values of a suite of 35 universally conserved
single copy genes (Orsi et al., 2015), per metagenome sample.
3.3 Factor Analysis
Q-mode Factor Analysis (QFA) was employed to simplify the ancient DNA dataset. Prior to the
factor analysis the DNA database was reduced to the 124 most abundant taxonomic units from a
total of 1,462 units identified by considering only those present in two or more samples with a
cumulative abundance higher than 0.5±0.1% (Table S1). The data was pretreated with a range-
normalization and run though the QFA with a VARIMAX rotation (Pisias et al., 2013). QFA
identified taxonomic groups that covary in our dataset and determined the minimum number of
components (i.e., factors) needed to explain a given fraction of the variance of the data set (Fig.
4; see supplementary materials). Each VARIMAX-rotated factor indicates an association of
taxonomic groups that covary (i.e., behave similarly amongst the samples). Taxonomic groups
that covary strongly within a factor will have high factor scores for that factor. We primarily
used dominant taxa with scores higher than 0.2 in a factor to interpret the plankton taxonomic
groups in that factor. The importance of a factor in any given sample is recorded by the factor
loading that we used to interpret the importance of that factor with depth/time downcore.
3.4 Foraminifera Counts
Samples for counting planktonic foraminifer *Globigerina falconensis* were wet-sieved over a 63-
μm screen. Typical planktonic foraminifer assemblages for the NE Arabian Sea were observed:
*Globigerinoides ruber, Neogloboquadrina dutertrei, Globigerina falconensis, Orbulina*
*universa, Globigerinoides sacculifer, Pulleniatina obliquiloculata, Globorotalia menardii*.
Counts of *Globigerina falconensis* were conducted on the size fraction >150 μm. We report
counts for the samples yielding >300 foraminifer individuals (see supplementary materials).
3.5 Harappan Sites
Archaeological site distribution provides an important line of evidence for social changes in the
Harappan domain (e.g., Possehl, 2000). We analyzed the redistribution of small (<20 ha), rural
vs. large (>20 ha), possibly urban sites on the G-H interfluve from the Early Harappan period,
through the Mature and Late periods to the post-Harappan Grey Ware culture (see supplementary
materials). Compared to settlements along the Indus and its tributaries that can be affected by
fluvial erosion (Giosan et al., 2012), the distribution of archaeological sites on G-H, where large
laterally-incising Himalayan rivers were absent during the Holocene, is probably more complete
and representative of their original distribution. To observe trends related to partial or complete
drying of the G-H system (Clift et al., 2012; Giosan et al., 2012; Singh et al., 2017), we divided
the settlements into upper and lower G-H sites located in the modern regions of Punjab and
Haryana in India, respectively Cholistan in Pakistan. For archaeological site locations and their
radiocarbon and/or archaeological ages we follow Giosan et al. (2012), using data from the
compilation by Gangal et al. (2001) with additions from regional gazetteers and surveys (Kumar,
2009; Mallah, 2010; Mughal, 1996 and 1997; Possehl, 1999; Wright et al., 2005).
4. Results
Exceptional preservation of organic matter in the OMZ (Altabet et al., 1995; Schulz et al., 2002)
allowed us to reconstruct the history of the planktonic communities based on their preserved
sedimentary DNA (see also Orsi et al., 2017). The factor analysis of the dominant DNA species
(Fig. 4) identified three significant factors that together explain 48% of the variability in the
dataset (see supplementary materials). Additional factors were excluded as they would have
increased the variability explained by an insignificant amount for each ($< 3\%$). We interpret
these factors as corresponding to the SST regime, nutrient availability, and sea level state,
respectively (Fig. 4). Factor 1 (Fig. 4c) explains 20% of the variability and is largely dominated
by radiolarians (*Polycystinea*) that prefer warmer sea surface conditions (e.g., Cortese and
Ablemann, 2002; Kamikuri et al, 2008). High scores for jellyfish (*Cnidaria*) that thrive in warm,
eutrophic waters (Purcell, 2005) also support interpreting Factor 1 as a proxy for a plankton
community adapted to high sea surface temperatures. A general increase of the Factor 1 loadings
since the early Holocene is in accordance with the $U^K_{37}$-reconstructed warming of Orsi et al.
(2017). During the Holocene, relatively colder conditions are evident in Factor 1 between ~4500
and 3000 years ago (Fig. 4) as previously detected in the higher resolution $U^K_{37}$ record from a
core located nearby on the Makran continental margin (Doose-Rolinski et al., 2001).
Factor 2 (Fig. 4b) explains 18% of the variability and is dominated by marine dinoflagellates
indicative of high nutrient, bloom conditions (e.g., Worden et al., 2015), flagellates (*Cercozoa*)
and fungi. Parasitic Alveolates (*Hematodinium* and *Syndiniales*) that typically appear during
blooms (Worden et al., 2015) are also important. Increased representation of chlorophyll
biosynthesis genes (Fig. 4) in sediment metagenomes (Orsi et al., 2017) indicate higher
productivity (Worden et al., 2015) during the Factor 2 peak. All these associations suggest that
Factor 2 is a nutrient-sensitive proxy with a peak that overlaps with the colder conditions
between ~4500 and 3000 years ago. The inland retreat of the Indus fluvial nutrient source as sea
level rose (see below) probably explains the asymmetry in Factor 2 that exhibits higher scores in
the early vs. late Holocene. Overall, Factors 1 and 2 suggests enhanced winter convective mixing
between ~4500 and 3000 years ago that brought colder, nutrient-rich waters to the surface.
Factor 3 (Fig. 4a) explains 10% variability and is dominated by a wide group of taxa. The main
identified contributors to Factor 3 include the coastal diatom *Eucampia* (Werner, 1977), the fish-
egg parasite dinoflagellate *Ichthyodinium*, also reported from coastal habitats (Shadrin, 2010),
and soil ciliates (*Colpodida*), which altogether suggest a nearshore environment with fluvial
inputs. The plankton community described by Factor 3 was dominant in the first half of the
Holocene and became scarce as the sea level rose (Camoin et al., 2004) and the Indus coast
retreated inland (Fig. 4) .
At a simpler ecological level, *Globigerina falconensis* is the dominant planktonic foraminifer in
the NE Arabian Sea under strong winter wind mixing conditions (Munz et al., 2015; Schulz et
al., 2002). Over the last six millennia, after the sea level approached the present level, and when
the plankton community was consistently outside the influence of coastal and fluvial processes,
*G. falconensis* shows a peak in relative abundance between ~4500 and 3000 years during the
cold reversal previously identified by the sedimentary ancient DNA (Fig. 4d). A similar peak in
*G. falconensis* was detected in core SO42-74KL from the western Arabian Sea upwelling area
(Schulz et al., 2002) suggesting that mixing occurred in the whole northern half of the Arabian
Sea (Fig. 4d).
5. Discussion
5.1 Winter Monsoon Variability in the Neoglacial
In concert with previous data from the northern Arabian Sea, our reconstructions suggest that
convective mixing conditions indicative of a stronger winter monsoon occurred between ~4,500
and 3,000 years ago. Another cold yet variable period in the northern Arabian Sea (Doose-
Rolinski et al., 2001) occurred after ~1500 years ago under strong winter monsoon mixing (Böll
et al., 2014; Munz et al., 2015) and is seen in the  *G. falconensis* record of Schulz et al. (2002)
but is not captured completely in our top-incomplete record. In accordance with modern
climatologies colder SSTs in the northern coastal Arabian Sea correspond to increased westerly
extratropical cyclones bringing winter rains as far as Baluchistan and the western Himalayas
(Fig. 3 and Suppl. Fig. 1). Pollen records offshore the Makran coast where rivers from
Baluchistan and ephemeral streams flood during winter (von Rad et al., 1999) indeed indicate
enhanced winter monsoon precipitation during between ~4,500 and 3,000 years ago (Ivory and
Lezine, 2009). Bulk chemistry of sediments from the same Makran core were used to infer
enhanced winter-monsoon conditions between 3900 and 3000 years ago (Lückge et al., 2001).
Although not specifically identified as winter precipitation, increased moisture between ~4,600
and 2,500 years ago was also documented immediately east of the Indus River mouths in the
now arid Rann of Kutch (Pillai et al., 2018).
In a comparison to published Holocene records (Fig. 5), two periods of weak interhemispheric
thermal gradient for areas poleward of 30°N and 30°S occurred on top of more gradual,
monotonic changes driven by the seasonality of insolation (Fig. 5e; Marcott et al., 2013;
Schneider et al., 2014). These intervals are coeval within the limits of age models with the strong
winter monsoon phases in the Arabian Sea (Fig. 5g) and southward swings of the Intertropical
Convergence Zone (ITCZ) in the western Atlantic Ocean (Fig. 5f; Haug et al., 2001). Occurring
when Neoglacial conditions became pervasive across the Northern Hemisphere (Solomina et al.,
2015), we identify the two late Holocene periods characterized by a series of low
interhemispheric thermal gradient intervals as the Early Neoglacial Anomalies (ENA) between
ca. 4,500 and 3,000 years ago and the Late Neoglacial Anomalies (LNA) after ~1,500,
respectively.
LNA includes well-known cold events such as the Little Ice Age (LIA), an episode of global
reach but particularly strong in the Northern Hemisphere (IPCC, 2103; Mann et al., 2009;
Neukom et al., 2014; PAGES 2k Consortium, 2013) and the preceding cold during the European
Migration Period (Büntgen et al., 2016). ENA is more enigmatic at this point. The high
resolution Cariaco ITCZ record showing successive southward excursions suggests a series of
"LIA-like events" (LIALE in short - a term proposed by Sirocko, 2015). Furthermore, a
dominantly negative phase of the North Atlantic Oscillation – NAO (Fig. 5b; Olsen et al., 2012)
occurred during ENA, similar to synoptic conditions during LIA. This negative NAO phase was
concurrent with moderate increases in storminess in the Greenland Sea, as shown by sea-salt
sodium in the GISP2 core (O'Brien et al., 1995) and a cooling of the Iceland Basin and probably
the Nordic Seas (Orme et al., 2018). During both ENA and LNA the tropical North Atlantic was
remarkably quiescent in terms of hurricane activity (Fig. 5d), which appears to be the direct
result of the prevailing southward position of the ITCZ (Donnelly and Woodruff, 2007; van
Hengstum et al., 2016).
At mid latitudes, a southward position for the Westerlies wind belt, as expected during negative
NAO conditions, is supported at the western end of our domain of interest by well-defined
increases in spring floods in the Southern Alps (Fig. 5c) during both ENA and LNA (Wirth et al.,
2013). A higher precipitation-evaporation state in the northern Levant (Fig. 5h; Cheng et al.,
2015) and positive balances from lake isotope records in the Eastern Mediterranean (Fig. 5i;
Roberts et al., 2011), including lakes in Iran, occur further along the southward Westerlies
precipitation belt. The preferential southward track of the Westerlies during ENA and LNA is
also in agreement with a stronger Siberian Anticyclone, the dominant mode of winter and spring
climate in Eurasia, as interpreted from increases in the GISP2 non-sea-salt potassium (Fig. 5a).
At the Far East end of the Westerly Jet, support comes from dust reconstructions in the Sea of
Japan (Nagashima et al. 2013) and modeling (Kong et al., 2017), which suggest that the
Westerlies stayed preferentially further south in the late Holocene. As in modern climatologies,
this suite of paleorecords supports our interpretation that stronger winter monsoon winds during
ENA and LNA in the northernmost Arabian Sea, that ought to have driven more convective
mixing at our core site, were accompanied by increased precipitation penetration along the
Westerlies' path across the Iranian Plateau, Baluchistan and Makran to the western Himalayas.
Aridification after ca. 4200 years ago in a series of sensitive records from southern East Africa to
Australia (Berke et al., 2012; de Boer et al., 2014; Denniston et al., 2013; Li et al., 2018; Russell
et al., 2003; Schefuss et al., 2011; Wurtzel et al., 2018) argue for a narrowing of the ITCZ
migration belt during ENA within and around the Indian Ocean domain (Li et al., 2018).
In addition to its paleoclimatological value for the Harappan domain (see discussion below), a
more fundamental question emerges from our analysis: what triggered ENA and LNA? The
reduced influence of insolation on the ITCZ during the late Holocene (e.g., Haug et al., 2001;
Schneider et al., 2014) could have provided favorable conditions for internal modes of climate
variability, either tropical or polar, to become dominant (e.g., Wanner et al., 2008; Debret et al.,
2009; Thirumalai et al., 2018). In order to explain intervals of tropical instabilities that did not
extend over the entire Neoglacial various trigger mechanisms and/or coupling intensities
between climate subsystems could be invoked. For example, the weaker orbital forcing increased
the susceptibility of climate to volcanic and/or solar irradiance, which have been proposed to
explain decadal to centennial time events such as the Little Ice Age (e.g., IPCC, 2103; Mann et
al., 2009; McGregor et al., 2005; PAGES 2k Consortium. 2013). For the recently defined Late
Antique Little Ice Age between 536 to about 660 AD, a cluster of volcanic eruptions sustained
by ocean and sea-ice feedbacks and a solar minimum have been proposed as triggers (Büntgen et
al., 2016). However, during ENA the solar irradiance was unusually stable without prominent
minima (Stuiver and Braziunas, 1989; Steinhilber et al., 2012). The volcanic activity in the
northern hemisphere was also not particularly higher during ENA than after (Zielenski et al.,
1996) and it was matched by an equally active southern hemisphere volcanism (Castellano et al.,
2005). As previously suggested for the Little Ice Age (Dull et al. 2010; Nevle and Bird, 2008),
we speculate that mechanisms related to changes in landcover and possibly landuse could have
instead been involved in triggering ENA.
Biogeophysical effects of aerosol, albedo and evapotranspiration due to landcover changes were
previously shown to be able to modify the position of ITCZ and lead to significant large scale
geographic alterations in hydrology (e.g., Chung and Soden, 2017; Dallmeyer et al., 2017;
Devaraju et al. 2015; Kang et al., 2018; Sagoo and Storelvmo, 2017; Tierney et al., 2017).
Similarly, changes in tropical albedo and concurrent changes in regional atmospheric dust
emissions due to aridification during the Neoglacial could have affected the ITCZ.
Anthropogenic early land use changes could have also led to large scale biogeophysical impacts
(e.g., Smith et al., 2016). Such landcover- and landuse-driven changes were time-transgressive
across Asia and Africa (e.g., Lezine et al., 2017; Jung et al., 2004; Prasad and Enzel; 2006;
Shanahan et al., 2015; Tierney et al., 2017; Wang et al. 2010; Kaplan et al., 2011) and could
have led to a generalized instability of the global climate as it passed from the warmer Holocene
Thermal Maximum state to the cooler Neoglacial state. Therefore the instability seen during
ENA may reflect threshold behavior of the global climate system characterized by fluctuations or
flickering (Dakos et al., 2008; Thomas, 2016) or a combination of different mechanisms
affecting the coupling intensity between climate subsystems (Wirtz et al. 2010).
5.2 Climate Instability and the Harappan Metamorphosis
In contrast to other urban civilizations of the Bronze Age, such as Egypt and Mesopotamia,
Harappans did not employ canal irrigation to cope with the vagaries of river floods despite
probable knowledge about this agricultural technology through their western trade network (e.g.,
Ratnagar, 2004). Instead, they relied on a multiple cropping system that started to develop prior
to their urban rise (Madella and Fuller, 2006; Petrie et al., 2017) and integrated the winter crop
package imported from the Fertile Crescent (e.g., wheat, barley, peas, lentil) with local summer
crops (e.g., millets, sesame, limited rice). A diverse array of cropping practices using inundation
and/or dry agriculture that were probably supplemented by labor-intensive well irrigation was
employed across the Indus domain, dependent on the regional characteristics of seasonal rains
and river floods (e.g., Weber 2003; Pokharia et al. 2014; Petrie and Bates, 2017; Petrie et al.,
2017). The alluvial plains adjacent to the foothills of the Himalayas were probably the Harappan
region's most amenable to multiple crops using summer monsoon and WD rains directly or
redistributed via the perennial and/or ephemeral streams of the G-H interfluve. The
orographically-controlled stability and availability of multiple water sources that could be used
to mitigate climate risks probably made this area more attractive as the inundation agriculture
faltered along the Indus and its tributaries when the summer monsoon became more erratic.
Aridity intensified over most of the Indian subcontinent as the summer monsoon rains started to
decline after 5,000 years ago (Ponton et al., 2012; Prasad et al., 2014). The closest and most
detailed summer monsoon reconstruction to the Harappan domain shows a highly variable
multicentennial trend to drier conditions between ca. 4,300 and 3,300 years ago (Fig. 6a and 6b;
Kathayat et al., 2017). Thresholds in evaporation-precipitation affecting lakes on the upper G-H
interfluve occurred during the same period (Fig. 6c; Dixit et al., 2014). The flood regime
controlled by this variable and declining summer monsoon became more erratic and/or spatially
restricted (Giosan et al., 2012; Durcan et al., 2017) making inundation agriculture less
dependable. Whether fast or over generations, the bulk of Harappan settlements relocated toward
the Himalayan foothills on the plains of the upper G-H interfluve (see supplementary materials;
Possehl, 2002; Kenoyer, 1998; Wright, 2010; Madella and Fuller, 2006; Giosan et al., 2017).
Abandoned by Himalayan rivers since the early Holocene (Giosan et al., 2012; Clift et al., 2012;
Singh et al., 2017; Dave et al., 2018), this region between the Sutlej and Yamuna was watered by
orographically-enhanced rain feeding an intricate small river network (e.g., Yashpal et al., 1980;
van Dijk et al., 2016; Orengo and Petrie, 2017).
During the aridification process the number of large, urban-sized settlements on the G-H
interfluve decreased and the number of small settlements drastically expanded (Fig. 6e and 6d
respectively). The rivers on the G-H interfluve merged downstream to feed flows along the
Hakra into Cholistan, at least seasonally, until the latest Holocene (Giosan et al., 2012; Fig. 2).
Regardless if these settlements on the lower G-H interfluve were temporary and mobile (Petrie et
al., 2017) most of them were abandoned (Fig. 6d; see also supplementary materials) as the region
aridified, suggesting that flows became less reliable in this region. However, the dense stream
network on the upper G-H interfluve must have played an important role in more uniformly
watering that region, whether perennially or seasonally. Remarkably, Late Harappan settling did
not extend toward the northwest along the entire Himalayan piedmont despite the fact that this
region must have received orographically-enhanced rains too (Fig. 3 and Suppl. Fig. 1). One
possible reason is that interfluves between Indus tributaries (i.e., Sutlej, Beas, Ravi, Chenab,
Jhelum; Fig. 2) are not extensive. These Himalayan rivers are entrenched and collect flows inside
their wide valleys rather than supporting extensive interfluve stream networks (Giosan et al.,
473 2012).
Our winter monsoon reconstruction suggests that WD precipitation intensified during the time of
urban Harappan collapse (Fig. 6f). As the summer monsoon flickered and declined at the same
time, the classical push-pull model (e.g., Dorigo and Tobler, 1983; Ravenstein, 1885; 1889)
could help explain the Harappan migration. Push-pull factors induce people to migrate from
negatively affected regions to more favorable locations. Inundation agriculture along the summer
flood-deficient floodplains of the Indus and its tributaries became too risky, which pushed people
out, in the same time as the upper G-H region became increasingly attractive due to augmented
winter rain, which pulled migrants in. These winter rains would have supported traditional winter
crops like wheat and barley, while drought tolerant millets could still be grown in rotation during
the monsoon season. Diversification toward summer crops took place during the Mature
Harappan period, as the winter monsoon steadily increased, beginning around 4,500 years ago
(Fig. 6f), but a greater reliance on rain crops after the urban collapse implies that intense efforts
were made to adapt to hydroclimatic stress at the arid outer edge of the monsoonal rain belt
(Giosan et al., 2012; Madella and Fuller, 2006; Petrie and Bates, 2017; Wright et al., 2008). The
longevity of the Late Harappan settlements in this region may be due to a consistent availability
of multiple year-round sources of water. Summer monsoon remained strong enough locally due
to orographic rainfall, while winter precipitation increased during ENA and both these sources
provided relief from labor-intensive alternatives such as well irrigation. The decline in the winter
monsoon between 3300 and 3000 years ago (Fig. 6) at the end of ENA could have also played a
role in the demise of the rural late Harappans during that time as the first Iron Age culture (i.e.,
the Painted Grey Ware) established itself on the Ghaggar-Hakra interfluve.
The metamorphosis of Indus civilization remains an episode of great interest. The degradation of
cities and disintegration of supra-regional elements of the Indus cultural system such as its script
need not be sudden to be defined as a collapse. However, recent contributions of
geoarchaeological and settlement patterns studies, together with refinements in chronology,
require higher levels of sophistication for addressing links between climatic shifts and cultural
decline. While variation in coverage and imprecision in dating sites require further efforts (Petrie
et al., 2017), it remains clear that there were shifts in the distribution of population and the range
of site sizes, with decline in the size of the largest sites. The impacts of climatic shifts while
remarkable from recent chronological correlations (e.g., Katahayat et al 2017) must now be
assessed regionally through a nuanced appreciation of rainfall quantities as well as its seasonality
(e.g., Madella and Fuller, 2006; MacDonald, 2011; Petrie et al., 2017; Wright et al., 2008). How
precipitation was distributed seasonally would have affected the long-term stability and upstream
sources of the stream and river network (Giosan et al 2012; Singh et al 2017). Our study suggests
broad spatial and temporal patterns of variability for summer and winter precipitation across the
Harappan domain but the local hydroclimate aspects, as well as the role of seasonal gluts or
shortage of rain on river discharge need also to be considered. For example, did the increase in
winter rain during ENA lead to more snow accumulation in the Himalayas that affected the
frequency and magnitude of floods along the Indus and its tributaries? Or did settlements in
Kutch and Saurashtra, regions of relatively dense habitation during Late Harappan times, also
benefit from increases in winter rains despite the fact that modern climatologies suggest scarce
local precipitation?
Local reconstructions of seasonal hydroclimatic regimes would greatly enhance our ability to
understand social and economic choices made by Harappans. Attempts made to reconstruct WD
precipitation in the western Himalayas (e.g., Kotlia et al., 2017) are confounded by the dominant
summer monsoon (c.f., Kathayat et el., 2017). Developing local proxies based on summer vs.
winter crop remains may provide a more fruitful route for disentangling the sources of water in
the Harappan domain (e.g., Bates et al., 2017). The Indus civilization, especially in the northern
and eastern regions, had a broad choice of crops of both seasons. Mixed cropping may have
become increasingly important, including drought-tolerant, but less productive, summer millets
that suited weakening monsoon and winter cereals, including drought-tolerant barley, that were
aided by the heightened winter rains of Late Harappan era. Facilitated by this climatic
reorganization during ENA, the eastward shift in settlements, while it may have undermined the
pre-eminence of the largest urban centres like Harappa, can be seen as a strategic adjustment in
subsistence to the summer monsoon decline. Ultimately, ENA is a synoptic pattern that provides
a framework to address the role of climate in interacting with social dynamics at a scale larger
than the Indus domain. As such, if ENA affected human habitation of the entire eastern Northern
Hemisphere, and particularly in the Fertile Crescent and Iran that also depend on winter rains,
remains to be assessed.
6. Conclusions
To assess the role of winter precipitation in Harappan history, we reconstructed the Indian winter
monsoon over the last 6000 years using paleobiological records from the Arabian Sea. According
to modern climatologies, strong winter monsoon winds correspond to rains along a zonal swath
extending through the western Himalayas. Changes in the planktonic community structure
indicative of cool, productive waters highlight strong winter monsoon conditions between ca.
4,500 and 3,000 years ago, an interval spanning the transition from peak development of the
urban Harappan to the demise of its last rural elements. Inferred increases in winter rains during
this time were contemporaneous with the regionally documented decline in summer monsoon,
which has previously been interpreted as detrimental to the inundation agriculture practiced
along the Indus and its tributaries. We propose that the combined changes in summer and winter
monsoon hydroclimate triggered the metamorphosis of the urban Harappan civilization into a
rural society. A push-pull migration can better explain the relocation of Harappans from summer
flood-deficient river valleys to the Himalayan piedmont plains with augmented winter rains and
a greater reliance on rainfed crops. Two seasons of cultivation helped to spread risk and enhance
sustainability. Summer and winter orographic precipitation above and across the piedmont plains
fed a dense stream network supporting agriculture close to another millennium for the rural late
Harappans.
Previous reconstructions and our new monsoon record, in concert with other paleoclimate series
from the Northern Hemisphere and Tropics, display two late Holocene periods of generalized
climate instability: ENA between ca. 4,500 and 3,000 years ago and LNA after ~1,500 years ago.
The reduced influence of insolation during the late Holocene could have provided favorable
conditions for internal modes of climate variability, either tropical or polar, to become dominant
and lead to such instability intervals. Both ENA and LNA occurred during low interhemispheric
thermal gradients and dominantly negative phases of NAO characterized by more southward
swings of both the ITCZ and Westerlies belt at mid northern latitudes, reduced hurricane activity
and increases in high-latitude storminess in the Atlantic. The preferential southward track of the
Westerlies during ENA and LNA is supported by increased rains from WDs from the Levant into
Iran and Baluchistan, but a stronger Siberian Anticyclone and weaker winds along the northern
Westerly track as far east as the Sea of Japan. Susceptibility of climate to volcanic, solar
irradiance and/or landcover were proposed to explain LNA but we speculate that time-
transgressive changes in landcover across Asia and Africa could have been involved in triggering
ENA as it passed from the warmer Holocene Thermal Maximum state to the cooler Neoglacial
state.

Note Added in Proof:

During the review of our manuscript, a paper on a similar topic was accepted for discussion in
this journal (Giesche et al., 2018, in review). The authors comment on our work and we provide
a brief reply herein. Giesche et al. (2018, in review) used multi-species planktonic foraminifer
$\delta^{18}$O and $\delta^{13}$C from a core close to our site to infer a history of the Indian winter monsoon
between 4.5-3.0 ka BP that is different than what we propose. We suggest that ancient DNA and
% Globigerina falconensis proxies are better suited to reconstruct monsoon changes by providing
the right balance between planktonic whole-ecosystem change and proxy specificity,
respectively.

Data Availability
Data presented in the paper can be accessed in the supplementary materials. After publication the
data will also be uploaded to to the Woods Hole Open Access Server (FAIR-aligned data
repository).
Author Contributions
L.G. and P.D.C. collected the core. M.C. and C.W. measured and interpreted ancient DNA.
A.G.D. performed factor analysis. K.T. provided climatology. L.G., W.D.O., K.T. and D.Q.F.
interpreted the results with input from all authors. L.G. wrote the manuscript with input from all
authors.
Competing Interests
The authors declare that they have no conflict of interest.
Acknowledgements

This work was supported by the NSF OCE Grant #0634731 and internal WHOI funds to LG,
NSF MGG Grant #1357017 to MJLC, VG, and LG, and a C-DEBI grant #OCE-0939564 to
WDO. We thank the editor and reviewers for suggestions that improved the original manuscript.
Thanks go to Mary Carman for help with foraminifera, Lloyd Keigwin for discussions and
Pakistani and Indian colleagues who helped with acquiring and/or provided access to data
including Kavita Gangal, Ronojoy Adhikari, Ali Tabrez, and Asif Inam.

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

plain to the west and reached its urban peak (Mature Phase) between ca. 4,500 and 3,900 years
ago. The Harappans were agrarian but developed large, architecturally complex urban centers
and a sophisticated material culture coupled with a robust trade system. In contrast to the
neighboring hydraulic civilizations of Mesopotamia and Egypt, Harappans appear to have
invested less effort to control water resources by large-scale canal irrigation near cities but relied
primarily on fluvial inundation for winter crops and additionally on rain for summer crops.
Deurbanization ensued after approximately 3,900 years ago and was characterized by the
development of increasingly regional artefact styles and trading networks, as well as the
disappearance of the distinctive Harappan script. Some settlements exhibited continuity, albeit
with reduced size, whereas many riverine sites were abandoned, in particular along the Indus and
its tributaries. Between ca. 3,900 and 3,200 years ago, there was a proliferation of smaller,
village-type settlements, especially on the Ghaggar-Hakra interfluve. Socio-economic as well as
environmental hypotheses have been invoked to explain the collapse of urban Harappan society,
including foreign invasions, social instabilities, trade decline, climate deterioration, fluvial
dynamics, and human-induced environmental degradation.

The "climate-culture hypothesis", first clearly articulated by Singh (1971) and Singh et al. (1974)
based on pollen records from Rajasthan lakes, argues for climate variability at the vulnerable arid
outer edge of the monsoonal rain belt as a determining factor in Harappan cultural
transformations (Fig. 1 and 2; Suppl. Fig. 4). These reconstructions together with other early
paleoclimate forays in Rajasthan (see review of Madella and Fuller, 2006) proposed that
enhanced summer monsoon rains assisted the development of the urban Harappan but weakening
monsoon conditions after 4,200-3,800 years ago contributed to its collapse. In marine sediments,
planktonic oxygen isotope records in a core from the Makran continental margin were
interpreted to suggest a reduction in the Indus river discharge ca. 4,200 years ago (Staubwasser
et al., 2003). More recent work, proximal to the Harappan heartland, provides strong support for
this "climate-culture hypothesis" while emphasizing the complexity of both spatiotemporal
hydroclimate pattern and Harappan cultural responses. Paleohydrological records from lakes in
northern Rajasthan and Haryana show wetter conditions prevailing during the Early Harappan
phase, providing favorable climate conditions for urbanization (Dixit et al., 2018) and a distinct
weakening of summer monsoon around 4,100 years ago (Fig. 6c; Dixit et al., 2014). Another
summer monsoon reconstruction from Sahiya cave above the Himalayan piedmont (Fig. 6a and
6b; Kathayat et al., 2017) shows a pluvial optimum during most of the urban phase followed by

drying after 4,100 years ago. This high resolution speleothem-based reconstruction also reveals that the multicentennial trend to drier conditions between ca. 4,100 and 3,200 years ago was in fact highly variable at centennial scales.

Studies of fluvial dynamics on the Harappan territory are consistent with a dry late Holocene affecting the Harappan way of life. Landscape semi-fossilization along the Indus and its tributaries suggest that floods became erratic and less extensive making inundation agriculture unsustainable for the post-urban Harappans (Giosan et al., 2012). In contrast to Himalayan tributaries of the Indus, which incised their alluvial deposits in early-mid Holocene, the lack of wide entrenched valleys on the Ghaggar-Hakra interfluve indicates that large, glacier-fed rivers did not flow across this region during Harappan times. Geochemical fingerprinting of fluvial deposits on the lower and upper Ghaggar-Hakra interfluve (Clift et al., 2012 and Dave et al., 2018 respectively) showed that the capture of the Yamuna to the Ganges basin occurred prior to the Holocene. Similarly, abandonment and infilling of a large paleochannel demonstrates that the Sutlej River relocated to its present course away from the Ghaggar-Hakra interfluve by 8,000 years ago, well before Harappan established themselves in the region (Singh et al., 2018). However, widespread fluvial redistribution of sediment from the upper Ghaggar-Hakra interfluve (e.g., Saini et al., 2009; Singh et al., 2018) all the way down to the lower Hakra (Clift et al., 2012) and toward the Nara valley (Giosan et al., 2012) suggests that monsoon rains were able to sustain smaller streams through that time, but as the monsoon weakened, rivers gradually dried or became seasonal, affecting habitability along their course.

If the climatic trigger for the urban Harappan collapse was probably the decline of the summer monsoon, the agricultural Harappan economy showed a large degree of adaptation to water availability. The long-lived survival of Late Harappan cultures until ca. 3,200 years ago under a drier climate and less active fluvial network is the subject of the present study and further ongoing efforts (e.g., Kotlia et al., 2017; Petrie et al., 2017) that seek to understand the variability in hydroclimate and moisture sources across the Indus domain and how these relate to agricultural adaptations.

Figure Captions

Fig. 1. Physiography, winds and precipitation sources for the Harappan domain. The dominant source during summer monsoon is the Bay of Bengal while Western Disturbances provide the moisture during winter. The extent of the Indus basin and Ghaggar-Hakra (G-H) interfluve are shown with purple and brown masks, respectively. Locations for the cores discussed in the text are shown.

Fig. 2. Geographical regions and rivers of the Indus domain discussed in text.

Fig. 3. Modern seasonal climatology for South Asia. Average precipitation as well as wind direction and intensity for the summer (June-July-August or JJA) and winter (December-January-February or DJF) months are presented in the left and right panels, respectively. Note the differences in scales between panels for both rainfall and winds. Data used come from the ERA-40 reanalysis dataset (Uppala et al., 2005) for winds (averaged from 1958-2001) and the TRMM dataset (Huffman et al., 2007) for rainfall (averaged from 1998-2014). The white box encompasses the upper G-H interfluve.

Fig. 4. Holocene variability in plankton communities as reflected by their sedimentary DNA factor loadings (panels marked a through c) and winter mixing-sensitive % *G. falconensis* (panel marked d) in core Indus 11C in the NE Arabian Sea. Relative chlorophyll biosynthesis proteins abundances are also shown. Sea level points are from Camoin et al. (2004); SSTs are from Doose-Rolinski et al. (2001); and *G. falconensis* census from the NW Arabian Sea is from Schulz et al. (2002). Triangles show radiocarbon dates for core Indus 11C. The period corresponding to the Early Neoglacial Anomalies (ENA) is shaded in red hues.

Fig. 5. Northern Hemisphere hydroclimatic conditions since the middle Holocene. The period corresponding to the Early Neoglacial Anomalies (ENA) interval is shaded in red hues. From high to low (panels marked a trough i): (a) Greenland dust from non-sea-salt $K^+$ showing the strength of the Siberian Anticyclone (O'Brien et al., 1995); (b) NAO proxy reconstruction (Olsen et al., 2012) and (c) negative NAO-indicative floods in S Alps (Wirth et al., 2013); (d) grainsize-based hurricane reconstruction in the N Atlantic (van Hengstum et al., 2016); (e) interhemispheric temperature anomaly (Marcott et al., 2013); (f) ITCZ reconstruction at the Cariaco Basin (Haug et al., 2011); (g) winter monsoon ancient DNA-based reconstruction for the NE Arabian Sea (this study – in purple); (h) speleothem $\delta^{18}$O-based precipitation reconstruction for northern Levant (Cheng et al., 2015); and (i) stacked lake isotope records as a proxy precipitation-evaporation regimes over Middle East and Iran (Roberts et al., 2011).

Fig. 6. Monsoon hydroclimate changes since the middle Holocene and changes in settlement distribution on the Ghaggar-Hakra interfluve. From high to low (panels marked a trough f): (a) variability in summer monsoon calculated as 200-year window moving standard deviation of the

detrended monsoon record of Katahayat et al. (2017) and (b) the speleothem $\delta^{18}$O-based summer
monsoon reconstruction of Katahayat et al. (2017); (c) lacustrine gastropod $\delta^{18}$O-based summer
monsoon reconstruction (Dixit et al., 2014); (d and e) changes in the number of settlements on
the Ghaggar-Hakra interfluve as a function of size and location; and (f) winter monsoon ancient
DNA-based reconstruction for the NE Arabian Sea (this study – in purple). The period
corresponding to the Early Neoglacial Anomalies (ENA) is shaded in red hues and durations for
Early (E), Mature (M) and Late (L) Harappan phases are shown with dashed lines.
Fig. 1

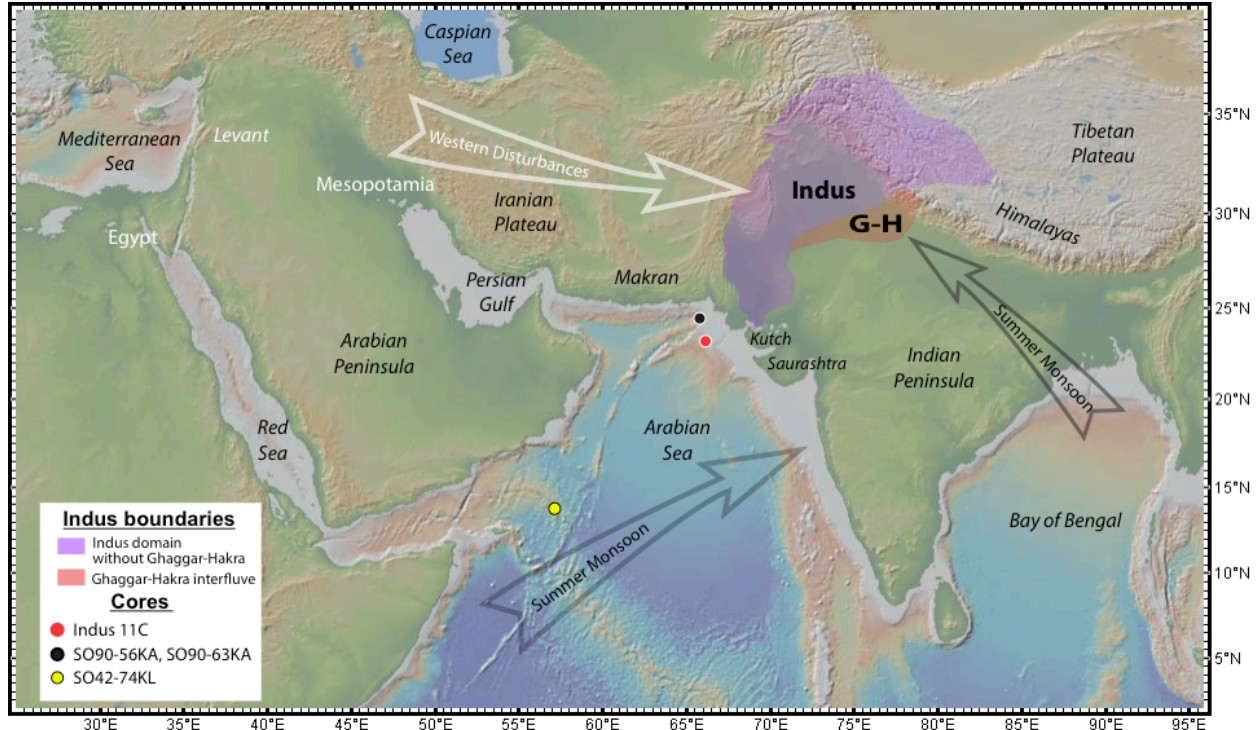


Fig. 2

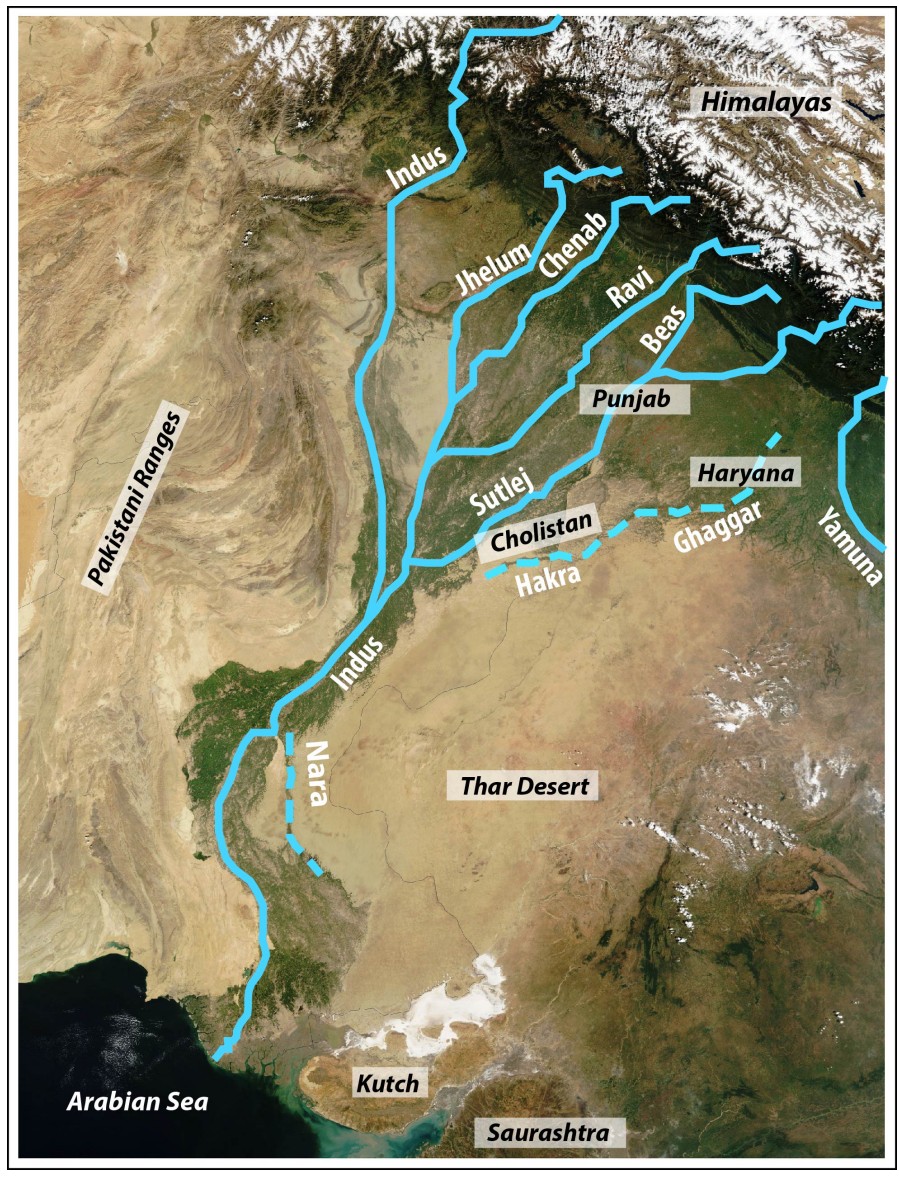

Fig. 2. Geographical regions and rivers of the Indus domain discussed in text.

Fig. 3.

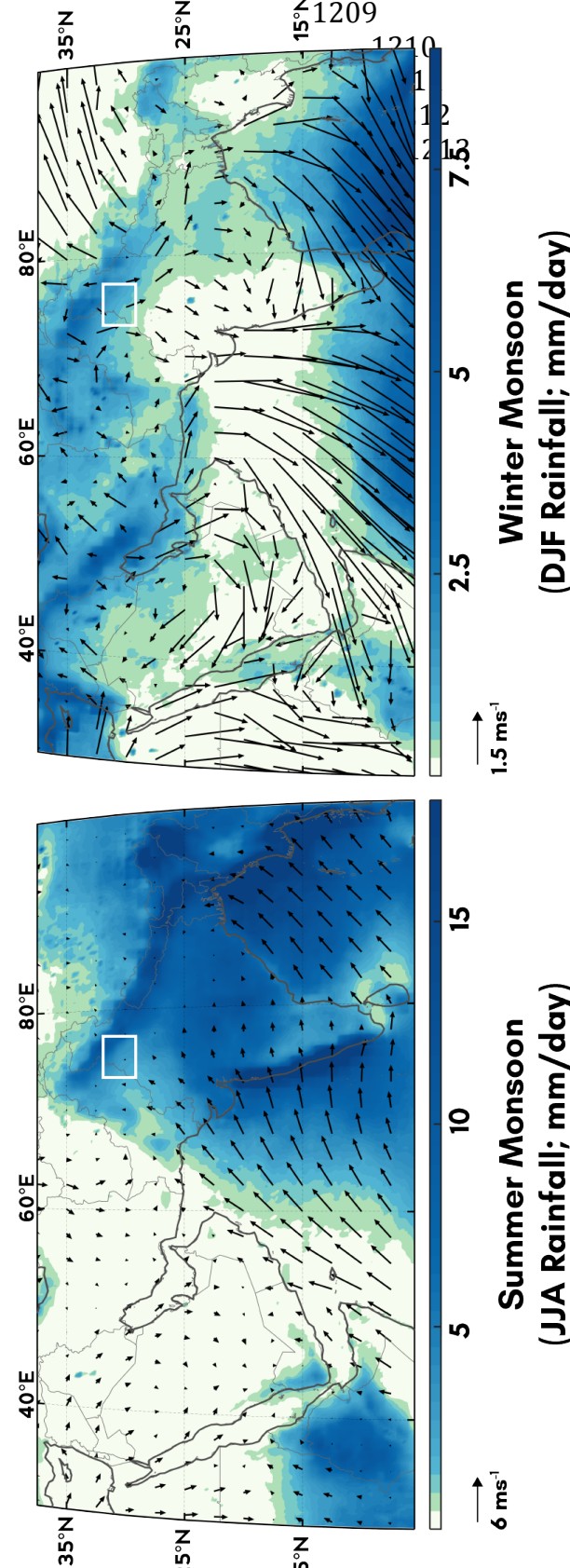

    Fig. 4

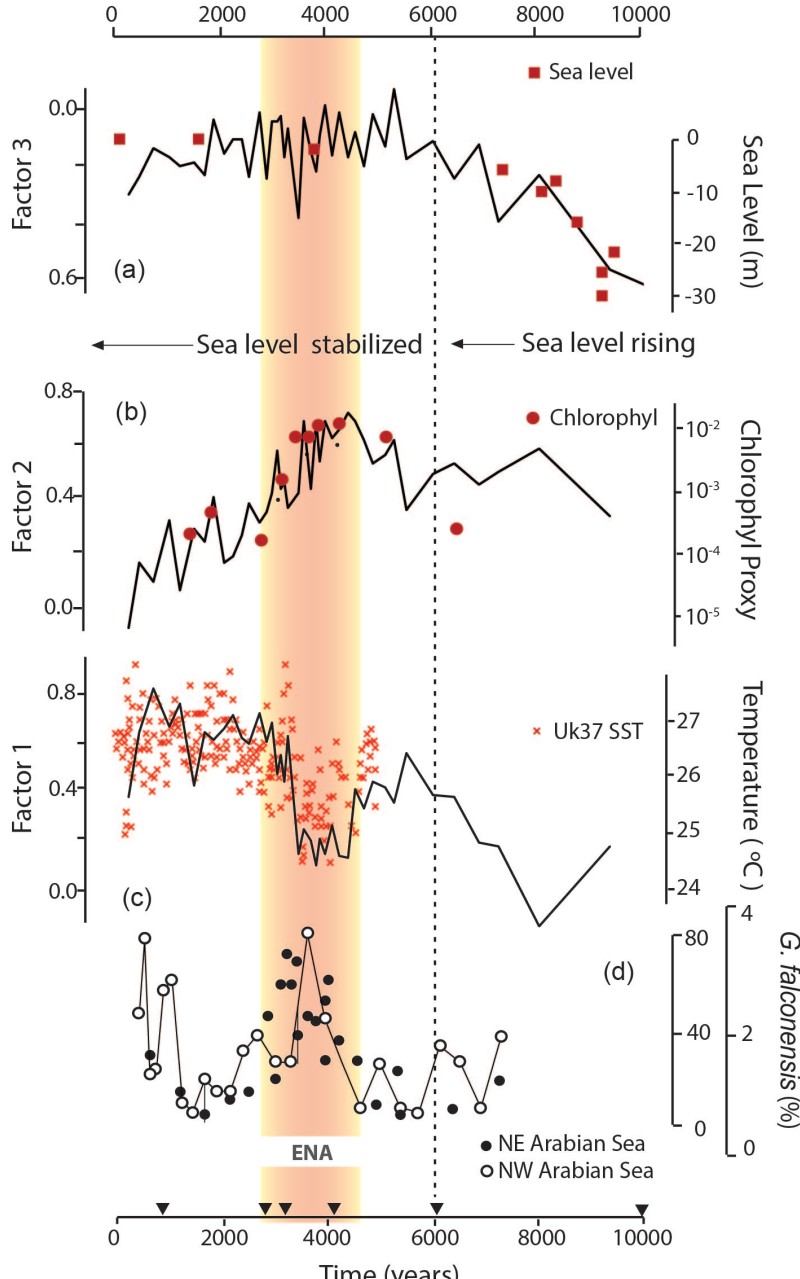

Fig. 5.

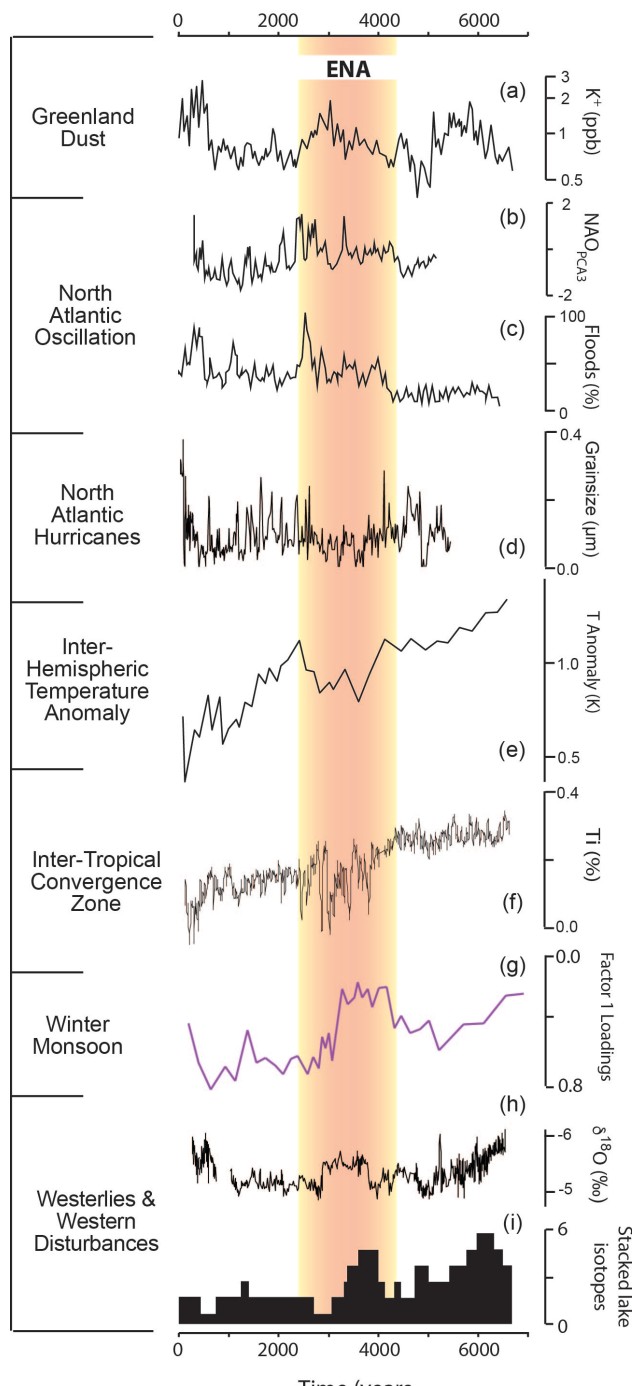


Fig. 6.

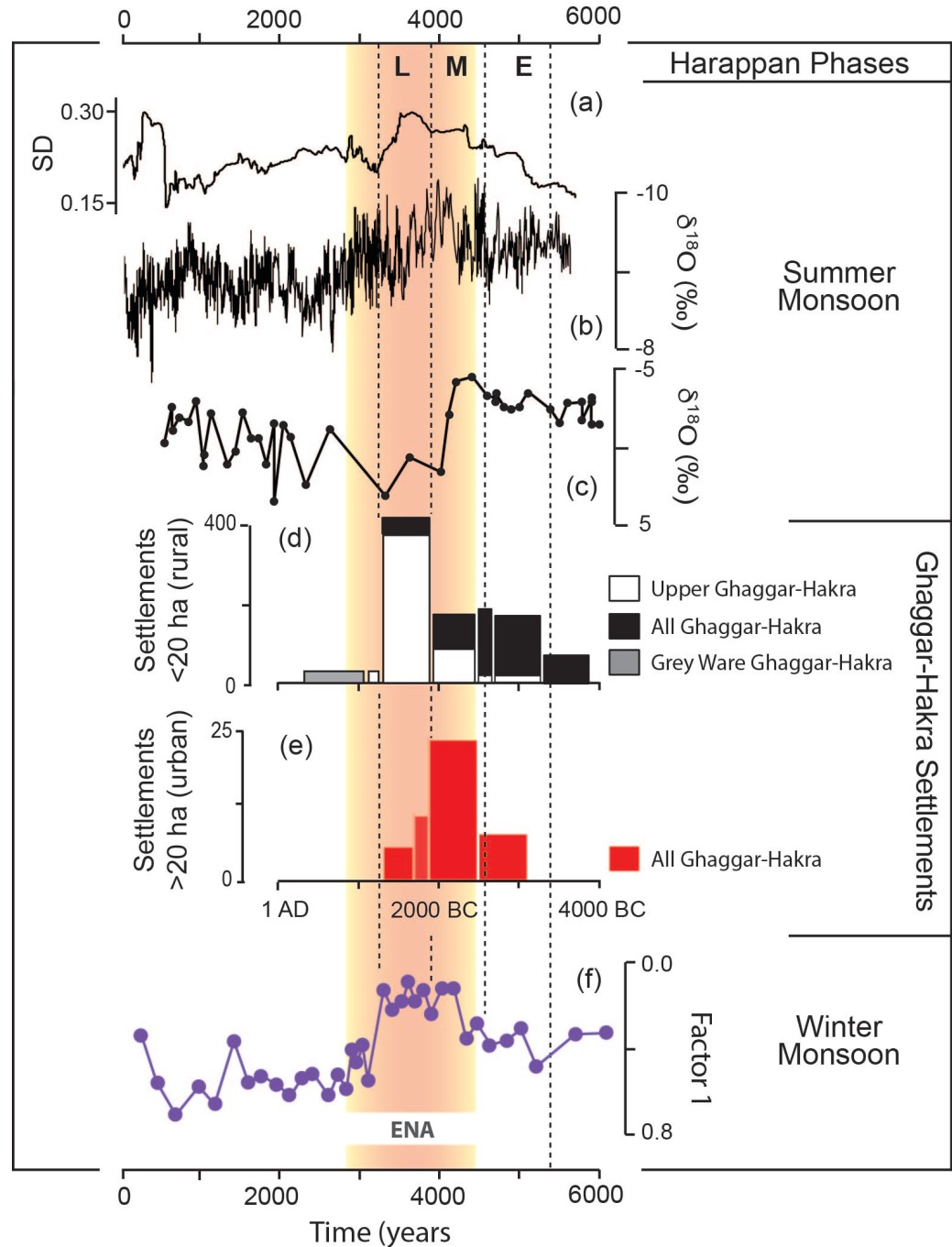
