# Peer review of "Neoglacial Climate Anomalies and the Harappan Metamorphosis Authors: Liviu Giosan1\*, William D. Orsi2,3, Marco Coolen4, Cornelia Wuchter4, Ann G. Dunlea1, Kaustubh Thirumalai5, Samuel E. Munoz1, Peter D. Clift6</s"

_Climate of the Past, 2018_

## Referee Comment (RC1) · Anonymous Referee #1 · 21 Jul 2018

The article, in reality, consists of two parts: the presentation of a new quantitative reconstruction of Indian monsoon winter precipitation and a discussion of the interlinkages between hydroclimatic changes (e.g. drought) and the collapse of the Harappan civilization. There is no problem in itself with this, although the fact that there are two separate "stories" from time to time makes it slightly more difficult to follow the article. The article is, in general, well written but additional polishing of the text would be preferable prior to publication. The text contains quite a number of typos (especially in the references). Moreover, especially the figures could be clearer and improved. As a minimum, all the graphs should be in colour to make them easier to read. The article is clearly suited for publication in Climate of the Past but first after a careful revision where the authors can consider my suggestions below.

[Figure]

Major comments:

I have no comments regarding the new Indian monsoon winter precipitation reconstruction. It is clearly an important palaeoclimatological contribution that in itself would merit publication in Climate of the Past. On the other hand, the general discussion about climatic–societal links in the past can clearly be improved. This field is nowadays large and the references provided are few and rather old. For example, I am missing the works by Carey (2012), McMichael (2012), Brooke (2014); Izdebski et al. (2015), d'Alpoim Guedes et al. (2016), Nelson et al. (2016), Ljungqvist (2017) and Haldon et al. (2018). The methodological and conceptional problems, and interdisciplinary challenges, connected with trying to link climatic changes with societal changes need to be discussed more.

I would also advice the authors to describe various aspects of the Harappan civilization more in detail on 1–2 pages. Without this information, it is difficult for a non-expert reader to assess if the links to drought that the authors make are plausible or not. I understand that an article of this kind cannot contain a full "handbook text" but more of an introduction to the Harappan civilization would nevertheless be helpful.

Finally, it would be helpful for the reader if the authors added a conclusion/summery of the new reconstruction at the end of the article. As it is now, the conclusion is mainly devoted to the collapse of the Harappan civilization.

Minor comments:

Lines 35–36: This sentence is a bit unclear. Do the authors mean that the Little Ice Age only occurred in the extra-tropical Northern Hemisphere? It was indeed global.

Lines 41—-42 and elsewhere: I am not entirely happy with the phrase "Early Neoglacial Anomaly" – the Neoglaciation started well before the event in question and it is thus not "early".

Line 43: Likely also in other parts of the world.

Lines 49–50 and elsewhere: Consider using "Holocene Thermal Maximum" instead of "Holocene Optimum".

Lines 56–57: Consider revision here. Archaeologists work with inferring societal changes, and their possible causal connections, in societies lacking written sources all the time.

Lines 59–60: Actually, our knowledge is in many cases rather good today so I would recommend to reformulate this sentence.

Line 313: "Boll" should be "Böll".

Line 332: Please, make it clear if this ENA is thought to extend all the way to the present.

Line 335: I would cite IPCC (2013) here rather than Mann et al. (2009).

Lines 336–339: How are these LIALE related to, or the same as, the (controversial) so-called "Bond events" detected for the North Atlantic region and elsewhere? I think this should be discussed here.

Line 370 onward: I am not entirely convinced that the impacts of solar forcing and volcanic forcing were necessarily smaller in a warmer world with stronger orbital forcing. The mean state of climate was different but not necessarily the centennial- to decadal-scale variations.

Lines 373–374: Again, you may cite IPCC (2013) here.

Fig. 1: Please, also insert in the legend directly in the figure what the coloured fields mean. Figs. 3–5: Please redraw the figures in colour and make them clearer. Now, both the graphs themselves and the text in them are not very distinct.

References:

Brooke, J.L., Climate Change and the Course of Global History: A Rough Journey

(New York: Cambridge University Press, 2014).

Carey, M., 'Climate and history: a critical review of historical climatology and climate change historiography,' Wiley Interdiscip. Rev. Clim. Change 3 (2012): 233–49.

d'Alpoim Guedes, J.A., et al., 'Twenty-first century approaches to ancient problems: Climate and society,' Proc. Natl. Acad. Sci. Unit. States Am. 113 (2016): 14483–91.

Haldon, J., et al., 'History meets palaeoscience: Consilience and collaboration in studying past societal responses to environmental change,' Proc. Natl. Acad. Sci. Unit. States Am. (2018): 3210–3218.

Izdebski, A., et al. (2015), 'Realising consilience: how better communication between archaeologists, historians and geoscientists can transform the study of past climate change in the Mediterranean,' Quat. Sci. Rev. 136 (2016): 5–22.

Ljungqvist, F.C., 'Human and societal dimensions of past climate change,' in C.L. Crumley et al. (eds.), Issues and Concepts in Historical Ecology: If the Past Teaches, What Does the Future Learn? (Cambridge 2017): 41–83.

Michael, A.J., 'Insights from past millennia into climatic impacts on human health and survival,' Proc. Natl. Acad. Sci. Unit. States Am. 109 (2012): 4730–7.

Nelson, M.C., et al., 'Climate challenges, vulnerabilities, and food security,' Proc. Natl. Acad. Sci. Unit. States Am. 113 (2016): 298–303.

---

## Referee Comment (RC2) · Anonymous Referee #2 · 3 Aug 2018

The paper presents a reconstruction of the Indian winter monsoon in the Arabian Sea for the last 6000 years based on paleobiological records with different complexity. Based on the analysis of sedimentary paleo-DNA and planktonic foraminifers the authors show that stronger winter monsoons occurred between ca. 4,500 and 3'000 years ago. They call this period Early Neoglacial Anomaly (ENA) and argue that this climate reorganization may have helped trigger the well-known metamorphosis of the urban Harappan civilization into a rural society. As a dynamical climatologist I could principally review the climatological part of the paper. I was not able to evaluate the methodological part of the sediment core analysis.

Overall the paper presents an interesting analysis on the activity of the Indian Winter Monsoon and its impacts on the Harappan civilization. I am personally cautious if a new term for a climate period is defined. Are the authors convinced the ENA is a global phenomenon with high significance? Is it not possible that similar climate periods marking the transition to the Neoglacial occurred even earlier and in other areas of the globe?

Specific comments:

Lines 34-36: Avoid creating the impression the Little Ice Age was not global. It was but, due to the inertial effect of the large ocean areas in the Southern Hemisphere, the cooling effect occurred later in this area (see Neukom et al., 2014, Nature Climate Change 4, 362-367).

Lines 48-49: If you call a climate period as an optimum, it has to be related to a certain state or process. Therefore, I recommend using the classical term "Holocene Thermal Maximum".

Lines 67-73: The Indian Winter Monsoon is, simply said, driven by the thermal contrast between the cold Asian continent and the adjacent warm oceans (see e.g. Trenberth et al. al., 2006, in: The Asian Monsoon, Springer; Wang and Chen, 2014, J. Climate 27, 2361-2374; Yancheva et al., 2007, Nature 445, doi:10.1038/nature05431).

Line 116: Dimri et al., 2015?

Line 220: Pisias et al, 2013 is not in reference list.

Line 313: Böll et al., 2014.

Line 332 and lines 364-369: I do not recommend introducing a new term called Late Neoglacial Anomaly (LNA). First of all, this period consists of two cooler (Migration Period, Little Ice Age) and two warmer periods (Medieval Climate Anomaly and Modern Warming). Second, the dynamical background differs clearly from the so-called ENA: Orbital forcing set the stage, Grand Solar Minima, volcanic events Interactive comment

and GHG forcing played a key role and, likely, internal variability had a significant influence (see Bradley et al., 2016, The Medieval Quiet Period. The Holocene, doi:10.1177/0959683615622552). Line 376: Büntgen et al., 2016 Lines 380-383: I agree, but we should not forget mentioning internal variability! Lines 431-432, 437: You mention several local names. I ask me if you should also add a Figure with a local map?

Lines 480-485: This is a very important question. I am asking me whether or not literature about this phenomenon is available?

Figure 1: I am not happy with the direction of the arrow marking the Summer Monsoon. Look at you Figure 2 A or consult Figure 1 in Chen et al., 2008, Quaternary Science Reviews 27, 351-364. Why did you not insert an arrow for Winter Monsoon?

Figures 2-5: In my opinion, for a better oprientation, it would make sense to denominate the Figures 2-5 with letters A,B, C etc.

Abbreviations: The paper contains numerous abbreviations. It would possibly make sense adding a list of abbreviations at the end of the paper text.

CPD

---

## Referee Comment (RC3) · Anonymous Referee #3 · 9 Aug 2018

Comments on Giosan et al "Neoglacial climate anomalies and the Harappan Metamorphosis"

This manuscript presents novel proxies for Indian winter monsoon variation form a core in the northeastern Arabian Sea and suggests that the intensified winter monsoon would contribute to the metamorphosis of the Harappan civilization from urban to rural society. The causal relationship between climate change and civilization has always been a question at debate due to lack of robust evidence. The variation in the winter monsoon and the distribution of the Harappan civilization archaeological sites in this paper is a great effort to answer this question. I support to be published the paper. However, there are some questions that the authors should address in next round of revision.

[Figure]

Major comments: 1. The manuscript is not written in a very clear way, which make readers hard to follow what the authors said. For example, in figures 3-5, all the curves should be marked by such as a, b, c, etc., and in the text it is easy to cite such as "Fig. 3c" to indicate the exact curve, but not such as (Fig. 5; Dixit et al., 2014) in Line 425. 2. The reference list should be carefully checked. Almost all references have some format problems or mistakes. For examples, Lines 516-518, use pp. to indicate pages, Lines 519-521 the pages are used "959-962". Also, the authors cited many published records in discussing Figure 3, but not showing any of them in the reference list. Readers and reviewers do not know what the authors discuss and compare when reading Line 259 to 322. Please check all the references in the References 3. It seems to me that the authors overinterpreted the records, though the proxies for winter monsoon variation is reasonably sounding. For example, the authors stated that the core top missed (Line 161~162). However, the authors put much effort in discussion on LNA (Line 335-345) while not showing any records from this core. Actually, the Factor 1 data do show many data points since 2000 years, which does not show the LNA though the authors claimed that Factor 1 reflects temperature change. Please explain why. 4. I am not convinced that changes in land cover and land use would affect movement of ITCZ. Please explain in detail. Does the authors mean the regions affected by heavy rains, which is not necessary the ITCZ? 5. The authors raised a very important question at the beginning in Introduction, "Moreover, our knowledge of temporal and spatial climatic patterns remains too restricted to fully address social dynamics" (Line 59~60). However, I did not see the authors address this question later in text other than discuss a little bit on interhemispheric temperature gradients. 6. The authors put much effort on distribution of Harappan sites, but did not show in any figures. It would be easier for readers to have a figures showing the distribution of the sites.

Minor comments: 1. Why the numbers of data vary so much for Factor 1, 2 and 3? Please clarify in the text. 2. Affiliations: should be consistent for all addresses. Some list to department, while others only list the university. 3. Introduction: The logic in Introduction is not clear. Please revise following clear logic. 4. Abstract: The Abstract

is not clear. For example, Line 32~34. 5. The authors did not label various panels in figures clearly, which makes reading difficult. Please label the panels and cite in the text. 6. The temporal resolution for samples should be clearly addressed. 7. What is Calib 7.129? (Line 159) 8. Line 272, should use "cal years BP" or "years" without "BP". Please check the whole text. 9. Line 312, should be "3,000 years ago" 10. Figure 1: Please check the arrow of "summer monsoon". The direction should be wrong. 11. Figure 3-5 quality is not high. Pleas improve them. 12. Figure caption. Figure 1, there are three colors in the figure 1 but not two. Please change the figure or the caption; Figure 4, Line 921, better to give the full name of "ENA". Figure 5, Line 943, change "of Dixit et al. (2014)" to "(Dixit et al., 2014)"

---

## Author Comment (AC1) · 18 Aug 2018

**Response to Reviewer 1:**

The article, in reality, consists of two parts: the presentation of a new quantitative reconstruction of Indian monsoon winter precipitation and a discussion of the interlinkages between hydroclimatic changes (e.g. drought) and the collapse of the Harappan civilization. There is no problem in itself with this, although the fact that there are two separate "stories" from time to time makes it slightly more difficult to follow the article. The article is, in general, well written but additional polishing of the text would be preferable prior to publication. The text contains quite a number of typos (especially in the references).

We are thankful for the reviewer's appreciation and suggestions. Typos are addressed.

Moreover, especially the figures could be clearer and improved. As a minimum, all the graphs should be in colour to make them easier to read.

We adopted a philosophy of minimal use of color to highlight the important points of each figure. However, we made a few changes that address the reviewer's point and increase readability: (1) we highlighted ENA in color; (2) we increased the visibility of records developed for this study by changing their color to distinguish them from other records used for comparison, and (3) colored some of the archaeological records that otherwise had a high potential to lead to confusions. See modified figures and captions at the end of this response.

The article is clearly suited for publication in Climate of the Past but first after a careful revision where the authors can consider my suggestions below. I have no comments regarding the new Indian monsoon winter precipitation reconstruction. It is clearly an important palaeoclimatological contribution that in itself would merit publication in Climate of the Past. On the other hand, the general discussion about climatic–societal links in the past can clearly be improved. This field is nowadays large and the references provided are few and rather old. For example, I am missing the works by Carey (2012), McMichael (2012), Brooke (2014); Izdebski et al. (2015), d'Alpoim Guedes et al. (2016), Nelson et al. (2016), Ljungqvist (2017) and Haldon et al. (2018). The methodological and conceptional problems, and interdisciplinary challenges, connected with trying to link climatic changes with societal changes need to be discussed more.

It was not our intention to expand the discussion of climate-society interactions but see no harm in adding a sentence to that regard with the series of references suggested. Indeed, these references that address largely the historical period cover a lot of ground especially due to availability of contemporaneous documents. The situation is a bit different for pre-historical cultures, and especially for the Indus, that do not benefit from written sources.

Modified section now reads: *"Moreover, our knowledge of temporal and spatial climatic patterns remains too restricted, especially deeper in time, to fully address social dynamics. Significant progress in addressing this problem have been made especially for*

*historical intervals (e.g., Carey, 2012; McMichael, 2012; Brooke, 2014; Izdebski et al., 2015; d'Alpoim Guedes et al., 2016; Nelson et al., 2016; Ljungqvist, 2017; Haldon et al., 2018). Still, the coalescence of migration phenomena, profound cultural transformations and/or collapse of prehistorical societies regardless of geographical and cultural boundaries during certain time periods characterized by climatic anomalies, events or regime shifts suggests that large scale climate variability may be involved (e.g., Donges et al., 2015 and references therein)."*

I would also advise the authors to describe various aspects of the Harappan civilization more in detail on 1–2 pages. Without this information, it is difficult for a non-expert reader to assess if the links to drought that the authors make are plausible or not. I understand that an article of this kind cannot contain a full "handbook text" but more of an introduction to the Harappan civilization would nevertheless be helpful.

We did present the basics for this in the original version and feel that expanding would make the paper much more complex and detract from its goal. There are excellent summaries already available for this topic that are cited in the text and can be accessed by the interested reader. One solution would be, if the editor agrees with that, to write a primer on the "Indus Civilization and Climate" as Text Box (treated similarly to a figure).

Finally, it would be helpful for the reader if the authors added a conclusion/ summary of the new reconstruction at the end of the article. As it is now, the conclusion is mainly devoted to the collapse of the Harappan civilization.

An entire section (5.1.) in the subchapter 5 ("Discussion with Conclusions") is dedicated to the new reconstruction. The fact that it is followed by section 5.2 dedicated to the Harappan may give the impression communicated by the reviewer. We do not think that restructuring subchapter 5 would change much in the economy of the paper. But if the editor feels that a separate conclusion subchapter is need we can add that.

Lines 35–36: This sentence is a bit unclear. Do the authors mean that the Little Ice Age only occurred in the extra-tropical Northern Hemisphere? It was indeed global.

LIA appears to have indeed been global, although this is not universally accepted. On the other hand LIA was particularly strong and prolonged in the Northern Hemisphere (NH), which indicates either a cause or a positive feedback in the NH as discussed in references cited. We added in the text that LIA has a global extent and cited appropriate references.

Modified section now reads: *"LNA includes well-known cold events such as the Little Ice Age (LIA), an episode of global reach but stronger and more extensive in the Northern Hemisphere (IPCC, 2103; Mann et al., 2009; Neukom et al., 2014) and the preceding cold during the European Migration Period (Büntgen et al., 2016)."*

Lines 41—-42 and elsewhere: I am not entirely happy with the phrase "Early Neoglacial Anomaly" – the Neoglaciation started well before the event in question and it is thus not "early".

The Neoglacial is not formally defined at a global scale as it is time-transgressive regionally. Instead we used the census approach of Solomina et al. (2015) where the Neoglacial became pervasive in the Northern Hemisphere since 4,500-4,400 years ago. ENA becomes manifest in most records around that time and extends for another ca. 1,500 years, which makes it early Neoglacial rather than late Neoglacial.

Line 43: Likely also in other parts of the world.

Indeed there are some records suggestive of ENA in some Southern Hemisphere (SH) locales where records of appropriate resolution exist and we added a sentence with references in that regard.

New text: *"Whether ENA was manifest in the Southern Hemisphere remains an open question. A south of the Equator record on the Congo Fan (Schefuss et al., 2005) as well as a continental margin reconstruction further south that integrates signals from the Orange River basin (Burdanowitz et al., 2018) both show a period of increased precipitation that is largely coeval with the dry Northern Hemisphere ENA."*

In the abstract we changed the phrasing to "accompanied by changes in wind and precipitation patterns that are particularly evident across the eastern Northern Hemisphere and Tropics" to leave open the problem to future studies in other regions.

Lines 49–50 and elsewhere: Consider using "Holocene Thermal Maximum" instead of "Holocene Optimum".

Changed accordingly.

Lines 56–57: Consider revision here. Archaeologists work with inferring societal changes, and their possible causal connections, in societies lacking written sources all the time.

Not clear what needs changing. We agree with the reviewer but that does not mean that such connections are not difficult to prove, especially at the scale of cultures and civilizations.

Lines 59–60: Actually, our knowledge is in many cases rather good today so I would recommend to reformulate this sentence.

We cannot agree with this point. In prehistory we lack the synoptic view afforded by modern or even historic climatic data to make such a claim yet.

Line 313: "Boll" should be "Böll".

Done.

Line 332: Please, make it clear if this ENA is thought to extend all the way to the present.

It was clearly defined just above that line: "…the Early Neoglacial Anomaly (ENA) between ca. 4,500 and 3,000 years ago…"

Line 335: I would cite IPCC (2013) here rather than Mann et al. (2009).

Added the suggested reference but also kept Mann et al. as it is a well-grounded, dedicated study of the problem.

Lines 336–339: How are these LIALE related to, or the same as, the (controversial) so-called "Bond events" detected for the North Atlantic region and elsewhere? I think this should be discussed here.

This is indeed a controversial issue that would be better discussed at large in a review-type context.

Line 370 onward: I am not entirely convinced that the impacts of solar forcing and volcanic forcing were necessarily smaller in a warmer world with stronger orbital forcing. The mean state of climate was different but not necessarily the centennial- to decadal-scale variations.

We agree with the reviewer and that is why we limited ourselves to examples based on cited literature. Some (e.g., Wirtz et al.) show increase or decrease in sub-orbital variability that is regionally organized. Testing how our suggested mechanism for ENA can be achieved in future modeling studies and is beyond the scope of our current study.

Lines 373–374: Again, you may cite IPCC (2013) here.

IPCC (2013) added.

Fig. 1: Please, also insert in the legend directly in the figure what the coloured fields mean.

Done.

[Figure]

Figs. 3–5: Please redraw the figures in colour and make them clearer. Now, both the graphs themselves and the text in them are not very distinct.

Some changes made. Please see explanations above and figures below.

[Figure]

[Figure]

[Figure]

---

## Author Comment (AC2) · 18 Aug 2018

**Response to Reviewer 3:**

This manuscript presents novel proxies for Indian winter monsoon variation form a core in the northeastern Arabian Sea and suggests that the intensified winter monsoon would contribute to the metamorphosis of the Harappan civilization from urban to rural society. The causal relationship between climate change and civilization has always been a question at debate due to lack of robust evidence. The variation in the winter monsoon and the distribution of the Harappan civilization archaeological sites in this paper is a great effort to answer this question. I support to be published the paper. However, there are some questions that the authors should address in next round of revision.

We appreciate the reviewer's comments and suggestions.

The manuscript is not written in a very clear way, which make readers hard to follow what the authors said. For example, in figures 3-5, all the curves should be marked by such as a, b, c, etc., and in the text it is easy to cite such as "Fig. 3c" to indicate the exact curve, but not such as (Fig. 5; Dixit et al., 2014) in Line 425.

We addressed this problem as suggested.

The reference list should be carefully checked. Almost all references have some format problems or mistakes. For examples, Lines 516-518, use pp. to indicate pages, Lines 519-521 the pages are used "959-962". Also, the authors cited many published records in discussing Figure 3, but not showing any of them in the reference list. Readers and reviewers do not know what the authors discuss and compare when reading Line 259 to 322. Please check all the references in the References

Done. However, we found no missing references that are cited in figure 3. If the reviewer identified such references we would appreciate if he/she can point them to us.

It seems to me that the authors overinterpreted the records, though the proxies for winter monsoon variation is reasonably sounding. For example, the authors stated that the core top missed (Line 161-162). However, the authors put much effort in discussion on LNA (Line 335-345) while not showing any records from this core. Actually, the Factor 1 data do show many data points since 2000 years, which does not show the LNA though the authors claimed that Factor 1 reflects temperature change. Please explain why.

We discuss LNA based on cores nearby where it is well attested – Doose-Rolinski et al. for temperature, Böll et al and Munz et al. for mixing. This was clear in our original text: *"Another cold yet variable period in the northern Arabian Sea (Doose-Rolinski et al., 2001) occurred after ~1500 years ago under strong winter monsoon mixing (Böll et al., 2014; Munz et al., 2015) and is seen in G. falconensis record of Schulz et al. (2002) but is not captured completely in our top-incomplete record."*

I am not convinced that changes in land cover and land use would affect movement of ITCZ. Please explain in detail. Does the authors mean the regions affected by heavy rains, which is not necessary the ITCZ?

Previous studies cited show that landcover and landuse can affect the ITCZ using both modeling and data – e.g., Chung and Soden, 2017; Devaraju et al. 2015; Kang et al., 2018; Smith et al., 2016.

The authors raised a very important question at the beginning in Introduction, "Moreover, our knowledge of temporal and spatial climatic patterns remains too restricted to fully address social dynamics" (Line 59-60). However, I did not see the authors address this question later in text other than discuss a little bit on interhemispheric temperature gradients.

We do show that ENA is detectable in the northern hemisphere (and now added references for a couple of cases in the southern hemisphere). This describes a novel temporal-spatial pattern affecting the winter rain in out region and suggests a "pull" factor for the Indus people's migration. We now added a sentence in the conclusions to link to this point made in the introduction.

It reads: *"Ultimately, ENA is a synoptic pattern that provides a framework to address the role of climate in interacting with social dynamics at a scale larger than the Indus domain. As such, if ENA affected human habitation of the entire eastern Northern Hemisphere, and particularly in the Fertile Crescent and Iran that also depend on winter rains, remains to be assessed."*

The authors put much effort on distribution of Harappan sites, but did not show in any figures. It would be easier for readers to have a figures showing the distribution of the sites.

We will add a supplementary figure addressing this.

Why the numbers of data vary so much for Factor 1, 2 and 3? Please clarify in the text.

The number of data for factors (black curves in Fig. 2) is the same for all factors (see also Suppl. Table 4). They are compared with other parameters (sea level, chlorophyll proxy, temperature) that each have their own resolution.

Affiliations: should be consistent for all addresses. Some list to department, while others only list the university.

Done.

Introduction: The logic in Introduction is not clear. Please revise following clear logic. Abstract: The Abstract is not clear. For example, Line 32-34.

These comments are unfortunately too vague. What one person might find as logic someone else might not. We would be happy to address them if clarified.

The authors did not label various panels in figures clearly, which makes reading difficult. Please label the panels and cite in the text.

Done.

The temporal resolution for samples should be clearly addressed.

Unclear what is requested. Temporal resolution is variable depending on sedimentation rates. Data is all documented in tables at the depth/time resolution available.

What is Calib 7.129? (Line 159)

This was a mistake – it is the Calib 7.1 calibration program – corrected and citation added.

Line 272, should use "cal years BP" or "years" without "BP". Please check the whole text.

We prefer to keep "years ago" throughout – now changed.

Line 312, should be "3,000 years ago"

Done.

Figure 1: Please check the arrow of "summer monsoon". The direction should be wrong.

The arrow indicates the direction of moisture reaching the area of interest during summer monsoon, which is from the Bay of Bengal. That is now made clearer in the caption.

Caption reads: *Fig. 1. Physiography and precipitation sources for the Harappan domain. The dominant source during summer monsoon is the Bay of Bengal while western disturbances provide the moisture during winter. The extent of the Indus basin and Ghaggar-Hakra (G-H) interfluve are shown with purple and brown masks, respectively. Locations for the cores discussed in the text are shown.*

Figure 3-5 quality is not high. Please improve them.

If the reviewer refers to the resolution of figures this will be improved in the final version. The submitted figures for the review stage have downgraded resolution to get the manuscript at a manageable size. We also implemented some color changes to increase readability.

Figure caption. Figure 1, there are three colors in the figure 1 but not two. Please change the figure or the caption; Figure 4, Line 921, better to give the full name of "ENA". Figure 5, Line 943, change "of Dixit et al. (2014)" to "(Dixit et al., 2014)"

Done.

---

## Author Comment (AC3) · 18 Aug 2018

**Response to Reviewer 2:**

The paper presents a reconstruction of the Indian winter monsoon in the Arabian Sea for the last 6000 years based on paleobiological records with different complexity. Based on the analysis of sedimentary paleo-DNA and planktonic foraminifers the authors show that stronger winter monsoons occurred between ca. 4,500 and 3,000 years ago. They call this period Early Neoglacial Anomaly (ENA) and argue that this climate reorganization may have helped trigger the well-known metamorphosis of the urban Harappan civilization into a rural society. As a dynamical climatologist I could principally review the climatological part of the paper. I was not able to evaluate the methodological part of the sediment core analysis. Overall the paper presents an interesting analysis on the activity of the Indian Winter Monsoon and its impacts on the Harappan civilization.

We thank the reviewer for his/her overall positive judgement on our paper and suggestions.

I am personally cautious if a new term for a climate period is defined. Are the authors convinced the ENA is a global phenomenon with high significance? Is it not possible that similar climate periods marking the transition to the Neoglacial occurred even earlier and in other areas of the globe?

ENA is evident in many records (presented initially in our paper, in others that we added now and even more others that are cited). It does not need to have global extent although one may be detected in the future (similar to the initial description of the LIA as Western European event). However, the fact that records of interhemispheric temperature gradient document ENA, it is a good indication that it may have had a global effect. It is also not necessary to have strict time bounds either – for example in the southern hemisphere (see Neukom et al. paper suggested by reviewer) LIA would have been shorter and lagged to the NH definition if first discovered and defined there.

Lines 34-36: Avoid creating the impression the Little Ice Age was not global. It was but, due to the inertial effect of the large ocean areas in the Southern Hemisphere, the cooling effect occurred later in this area (see Neukom et al., 2014, Nature Climate Change 4, 362-367).

LIA appears to have indeed been global, although this is not universally accepted. On the other hand LIA was particularly strong and prolonged in the Northern Hemisphere (NH), which indicates either a cause or a positive feedback in the NH as discussed in references cited. We added in the text that LIA has a global extent and cited appropriate references. Reference suggested now added.

Lines 48-49: If you call a climate period as an optimum, it has to be related to a certain state or process. Therefore, I recommend using the classical term "Holocene Thermal Maximum".

Done.

Lines 67-73: The Indian Winter Monsoon is, simply said, driven by the thermal contrast between the cold Asian continent and the adjacent warm oceans (see e.g. Trenberth et al. al., 2006, in: The Asian Monsoon, Springer; Wang and Chen, 2014, J. Climate 27, 2361-2374; Yancheva et al., 2007, Nature 445).

While references listed mainly concern the East Asian Winter Monsoon, a phenomenon with different dynamics and magnitude compared with the Indian Winter Monsoon (see e.g. Wang et al., 2003, Marine Geology or Munz et al., 2017), we have updated our text to incorporate the fact that the initial driver for the winter monsoon disturbances is potentially the thermal contrast between the Asian continent and the Indian Ocean (Dimri et al. 2016).

Line 116: Dimri et al., 2015?

Fixed.

Line 220: Pisias et al, 2013 is not in reference list.

Fixed.

Line 313: Böll et al., 2014.

Fixed.

Line 332 and lines 364-369: I do not recommend introducing a new term called Late Neoglacial Anomaly (LNA). First of all, this period consists of two cooler (Migration Period, Little Ice Age) and two warmer periods (Medieval Climate Anomaly and Modern Warming). Second, the dynamical background differs clearly from the so-called ENA: Orbital forcing set the stage, Grand Solar Minima, volcanic events and GHG forcing played a key role and, likely, internal variability had a significant influence (see Bradley et al., 2016, The Medieval Quiet Period. The Holocene, doi:10.1177/0959683615622552).

We see the reviewer's point. However, both ENA and LNA are composed of a series of anomalies (best expressed in the high resolution Cariaco ITCZ reconstruction but also in other records mentioned in the text - see figure 3) separated by a more quiescent interval. The problem then becomes the use of the singular form ("anomaly") that indeed does not reflect the above described situation. This is now addressed by changing to the use of plural form: ENA – Early Neoglacial Anomalies and LNA – Late Neoglacial Anomalies. At this stage, we do not and cannot tackle the ultimate causes of each of these anomalies but only a possible mechanism of transmitting them at larger geographical scales (i.e., the inter-hemispheric thermal balance).

Line 376: Büntgen et al., 2016

Fixed.

Lines 380-383: I agree, but we should not forget mentioning internal variability!

Internal variability was and is mentioned right at the top of the paragraph: "…could have provided favorable conditions for internal modes of climate variability, either tropical or polar, to become dominant…"

Lines 431-432, 437: You mention several local names. I ask me if you should also add a Figure with a local map?

We added a supplementary figure with a map of geographical names.

[Figure]

Lines 480-485: This is a very important question. I am asking me whether or not literature about this phenomenon is available?

We could not track such seasonality in literature. But a recently published paper (now cited) suggests increased rain during ENA in the Kutch/Saurashtra region.

Figure 1: I am not happy with the direction of the arrow marking the Summer Monsoon. Look at your Figure 2 A or consult Figure 1 in Chen et al., 2008, Quaternary Science Reviews 27, 351-364. Why did you not insert an arrow for Winter Monsoon?

Fig. 1 shows direction of the dominant moisture sources during summer and winter for the Harappan domain, which are not necessarily monsoons directions. Fig. 2 shows that instead. We clarified this more in the caption.

Caption now reads: *Fig. 1. Physiography and precipitation sources for the Harappan domain. The dominant source during summer monsoon is the Bay of Bengal while western disturbances provide the moisture during winter. The extent of the Indus basin and Ghaggar-Hakra (G-H) interfluve are shown with purple and brown masks, respectively. Locations for the cores discussed in the text are shown.*

Figures 2-5: In my opinion, for a better orientation, it would make sense to denominate the Figures 2-5 with letters A, B, C etc.

Fixed.

Abbreviations: The paper contains numerous abbreviations. It would possibly make sense adding a list of abbreviations at the end of the paper text.

We can certainly do that if the journal would accommodate it but wait for the editor's decision on this point.

---

## Referee Report (RR1)

I am very pleased with the way the authors have addressed the previous concern raised by me and the other reviewers. Thus, I recommend that the article is accepted now pending very minor corrections.

I am overall pleased with the revision and especially with the addition of the Text Box. I would, however, suggest to include the new Supplementary map showing the locations in the main text.

Moreover, instead of only citing for the Little Ice Age IPCC (2013), Mann et al. (2009), and Neukom et al. (2014) I would also recommend to cite:

PAGES 2k Consortium. 2013: Continental-scale temperature variability during the past two millennia. Nature Geoscience, 6: 339–346.

The correct title of Ljungqvist (2017) is: Issues and Concepts in Historical Ecology: The Past and Future of Landscapes and Regions

---

## Author Response (AR2)

**Dear Associate Editor:**

We appreciate your assessment and the reviewers' final suggestions. Herein we discuss them point by point to explain how we addressed them in the submitted revised manuscript.

**Reviewer 1:**

I am very pleased with the way the authors have addressed the previous concern raised by me and the other reviewers. Thus, I recommend that the article is accepted now pending very minor corrections.

I am overall pleased with the revision and especially with the addition of the Text Box. I would, however, suggest to include the new Supplementary map showing the locations in the main text.

**Done.**

Moreover, instead of only citing for the Little Ice Age IPCC (2013), Mann et al. (2009), and Neukom et al. (2014) I would also recommend to in addition cite:

PAGES 2k Consortium. 2013: Continental-scale temperature variability during the past two millennia. Nature Geoscience, 6: 339–346.

**Done.**

The correct title of Ljungqvist (2017) is: Issues and Concepts in Historical Ecology: The Past and Future of Landscapes and Regions

**Corrected.**

**Reviewer 2:**

**General remarks**

Compared to its first version and based on the reviewers comments, the paper has heavily improved. I am still not very happy with the new labels ENA and LNA, but can live with it if it is published in this form.

**Specific remarks**

**Lines 359-360 and 372:**

I agree the tracks of the Westerlies lie more south in case of negative NAO indices. But, in this case, the subpolar North Atlantic area is rather warm, not cool (see e.g. Visbeck et al., PNAS Nov. 6/2001). In addition, I am not convinced that the storminess increased. We thank the reviewer and clarify these points. Citing Orme et al. 2018:

"During a negative NAO, over sub-annual timescales, the response to air-sea heat fluxes and wind-driven Ekman transport is warming in a zonal band spanning the North Atlantic north of 45N (Kushnir, 1994; Seager et al., 2000; Visbeck et al., 2003). However, over multi-annual to decadal/centennial timescales it is suggested that a negative) NAO causes decreased convective activity in the Labrador Sea and weakening of the SPG and meridional overturning circulation, resulting in cooling)north of 55N (Eden and Jung, 2001; Visbeck et al., 2003; H€akkinen and Rhines, 2004; Latif et al., 2006). During a negative NAO strengthened northerly winds to the east of Greenland can reinforce the East Greenland Current (EGC) and increase the export of sea ice and freshwater from the Arctic to the North Atlantic, a scenario which in the twentieth century caused 'Great Salinity Anomalies' (Dickson et al., 1996: Delworth et al., 1997; Belkin et al., 1998; Blindheim et al., 2000; Ionita et al., 2016). Similar episodes have been identified over decadal-centennial timescales in model and paleoclimate analyses (Delworth et al., 1997; Renssen et al., 2005; Sicre et al., 2008; Ran et al., 2011)." Also increased storminess in the enhanced storm intensity in the Greenland Sea is the common interpretation of non-sea-salt Na (Nesje et al., 2008; Giraudeau et al., 2010; Trouet et al., 2012). Decrease in storminess at mid latitudes does not negate that. Accordingly we changed the text to clarify these points. Text reads now:

"This negative NAO phase was concurrent with moderate increases in storminess in the Greenland Sea, as shown by sea-salt sodium in the GISP2 core (O'Brien et al., 1995) and a cooling of the Iceland Basin and probably the Nordic Seas (Orme et al., 2018)."

Figure 1:

I am still not happy with the arrow for the Indian Summer Monsoon (ISM) on Figure 1. Based on the discussion with two specialists and based on paper studies it is clear that the source area of the ISM also includes the Arabian Sea (see also Figure 2).

We added an additional arrow for the summer monsoon that shows its average direction.

Line 1146: Marcott

Corrected.

**1 Neoglacial Climate Anomalies and the Harappan Metamorphosis**

- 2 3 Authors:
- 4
- 5 Liviu Giosan1\*, William D. Orsi2,3, Marco Coolen4, Cornelia Wuchter4,
- 6 Ann G. Dunlea1, Kaustubh Thirumalai5, Samuel E. Munoz1, Peter D. Clift6,
- 7 Jeffrey P. Donnelly1, Valier Galy7, Dorian Q. Fuller8
- 8
- 9
- 10 Affiliations:
- 11

[revised manuscript text omitted]

224                                                                                     |  <li>5. Discussion</li> <li>5.1 Winter Monsoon Variability in the Neoglacial</li> <li>In concert with previous data from the northern Arabian Sea, our reconstructions suggest that</li>                                                                                                                                                                                                                                                                                                                                                                                                                                                                                                                                                                                                                                                                                                                                                                                                                                                                                                                                                                                                                                                                                                                                                                                                              | Liviu Giosan 10/17/2018 1:41 PM
| 328

225                                                                              |  <li>5. Discussion</li> <li>5.1 Winter Monsoon Variability in the Neoglacial</li> <li>In concert with previous data from the northern Arabian Sea, our reconstructions suggest that convective mixing conditions indicative of a stronger winter monsoon occurred between ~4,500 and 2,000 warr area. Another cold yet wrights period in the partners Arabian Sea (Decce)</li>                                                                                                                                                                                                                                                                                                                                                                                                                                                                                                                                                                                                                                                                                                                                                                                                                                                                                                                                                                                                                        | Liviu Giosan 10/17/2018 1:41 PM
| 328

226                                                                       |  <li>5. Discussion</li> <li>5.1 Winter Monsoon Variability in the Neoglacial</li> <li>In concert with previous data from the northern Arabian Sea, our reconstructions suggest that convective mixing conditions indicative of a stronger winter monsoon occurred between ~4,500 and 3,000 years ago. Another cold yet variable period in the northern Arabian Sea (Doose-Balingli et al. 2001) accurred after. 1500 years ago under strong winter monsoon mixing (Döll)</li>                                                                                                                                                                                                                                                                                                                                                                                                                                                                                                                                                                                                                                                                                                                                                                                                                                                                                                                         | Liviu Giosan 10/17/2018 1:41 PM
| 328

[revised manuscript text omitted]

Liviu Giosan 10/17/2018 1:41 PM

**Liviu Giosan 10/17/2018 1:41 PM Deleted: 4b**

[revised manuscript text omitted]

Liviu Giosan 10/17/2018 1:41 PM Deleted: sulfate

Liviu Giosan 10/17/2018 1:41 PM Deleted: aNing Liviu Giosan 10/17/2018 1:41 PM Deleted: ,:

Liviu Giosan 10/17/2018 1:41 PM Deleted: D'Alpoim

- 725 2014.
- Debret, M., Sebag, D., Crosta, X., Massei, N., Petit, J.R., Chapron, E. and Bout-Roumazeilles,
   V.: Evidence from wavelet analysis for a mid-Holocene transition in global climate forcing,
   Quat. Sci. Rev., 28, 2675-2688, 2009.
- deMenocal PB: Cultural responses to climate change during the late Holocene, Science, 292,
   667–673, 2001.
- Denniston, R.F., Wyrwoll, K.H., Polyak, V.J., Brown, J.R., Asmerom, Y., Jr., A.D.W., Lapointe,
   Z., Ellerbroek, R., Barthelmes, M., and Cleary, D.: A Stalagmite record of Holocene
   Indonesian–Australian summer monsoon variability from the Australian tropics, Quat. Sci.
- Rev., 78, 155-168, 2013.
  Devaraju, N., Govindasamy B., and Angshuman M.: Effects of large-scale deforestation on
- precipitation in the monsoon regions: Remote versus local effects, Proc. Natl. Acad. Sci.
   India 112.11, 3257-3262, 2015.
- Dimri, A.P., Niyogi, D., Barros, A.P., Ridley, J., Mohanty, U.C., Yasunari, T., Sikka, D.R.:
  Western disturbances: a review. Rev. Geophys., 53, 225–246, 2015.
- Dimri, A. P.: Surface and upper air fields during extreme winter precipitation over the western
   Himalayas, Pure Appl. Geophys., 163, 1679–1698, 2006.
- Dixit, Y., Hodell, D.A., Petrie, C.A.: Abrupt weakening of the summer monsoon in northwest
   India ~4100 yr ago, Geology 42, 339–342, 2014.
- 744 Dixit, Y., Hodell, D.A., Giesche, A., Tandon, S.K., Gázquez, F., Saini, H.S., Skinner, L.C.,
- Mujtaba, S.A.I., Pawar, V., Singh, R.N., and Petrie, C. A. (2018). Intensified summer
  monsoon and the urbanization of Indus Civilization in northwest India, Scientific Reports,
  8(1), 4225.
- Donges, J.F., Donner, R., Marwan, N., Breitenbach, S.F., Rehfeld, K. and Kurths, J.:\_Non-linear
   regime shifts in Holocene Asian monsoon variability: potential impacts on cultural change
   and migratory patterns, Climate of the Past, 11, 709-741, 2015.
- Donnelly, J.P. and Woodruff, J.D.: Intense hurricane activity over the past 5,000 years controlled
   by El Niño and the West African monsoon, Nature, 447, 465-468, 2007.
- Doose-Rolinski, H., Rogalla, U., Scheeder, G., Lückge, A. and Rad, U.: High-resolution
   temperature and evaporation changes during the late Holocene in the northeastern Arabian
   Sea, Paleoceanography, 16, 358-367, 2001.
- Dorigo, G. and W. Tobler, W.: Push-pull migration laws, Ann. Assoc. Am. Geogr., 73, 1–17, 1983.
- Dull, R.A., Nevle, R.J., Woods, W.I., Bird, D.K., Avnery, S. and Denevan, W.M.: The
  Columbian encounter and the Little Ice Age: Abrupt land use change, fire, and greenhouse
  forcing, Ann. Assoc. Am. Geogr., 100, 755-771, 2010.
- Durcan, J.A., Thomas, D.S., Gupta, S., Pawar, V., Singh, R.N. and Petrie, C.A.: Holocene
  landscape dynamics in the Ghaggar-Hakra palaeochannel region at the northern edge of the
  Thar Desert, northwest India. Quat. Int., in press, 2017.
- 764 Edgar, R.C.: Search and clustering orders of magnitude faster than BLAST, Bioinformatics, 26,
   765 2460-2461, 2010.
- Find Find Find Structure
  Find Find Structure
  Find Find Structure
  Find Find Find Structure
  Find Find Find Structure
  Find Find Find Structure
  Find Structure</l

[revised manuscript text omitted]

Liviu Giosan 10/17/2018 1:41 PM Moved up [1]: Suppl. Fig. Liviu Giosan 10/17/2018 1:41 PM Deleted: 4).

Liviu Giosan 10/17/2018 1:41 PM Deleted: Figs. 3 and

Liviu Giosan 10/17/2018 1:41 PM Deleted: 5c Liviu Giosan 10/17/2018 1:41 PM Deleted: 5a Liviu Giosan 10/17/2018 1:41 PM Deleted: 5b

- drying after 4,100 years ago. This high resolution speleothem-based reconstruction also reveals that the multicentennial trend to drier conditions between ca. 4,100 and 3,200 years ago was in
- 1152 fact highly variable at centennial scales.
- 1153

1154 Studies of fluvial dynamics on the Harappan territory are consistent with a dry late Holocene 1155 affecting the Harappan way of life. Landscape semi-fossilization along the Indus and its 1156 tributaries suggest that floods became erratic and less extensive making inundation agriculture unsustainable for the post-urban Harappans (Giosan et al., 2012). In contrast to Himalayan 1157 1158 tributaries of the Indus, which incised their alluvial deposits in early-mid Holocene, the lack of 1159 wide entrenched valleys on the Ghaggar-Hakra interfluve indicates that large, glacier-fed rivers 1160 did not flow across this region during Harappan times. Geochemical fingerprinting of fluvial 1161 deposits on the lower and upper Ghaggar-Hakra interfluve (Clift et al., 2012 and Dave et al., 1162 2018 respectively) showed that the capture of the Yamuna to the Ganges basin occurred prior to 1163 the Holocene. Similarly, abandonment and infilling of a large paleochannel demonstrates that the 1164 Sutlej River relocated to its present course away from the Ghaggar-Hakra interfluve by 8,000 1165 years ago, well before Harappan established themselves in the region (Singh et al., 2018). 1166 However, widespread fluvial redistribution of sediment from the upper Ghaggar-Hakra interfluve (e.g., Saini et al., 2009; Singh et al., 2018) all the way down to the lower Hakra (Clift et al., 1167 2012) and toward the Nara valley (Giosan et al., 2012) suggests that monsoon rains were able to 1168 1169 sustain smaller streams through that time, but as the monsoon weakened, rivers gradually dried 1170 or became seasonal, affecting habitability along their course. 1171 1172 If the climatic trigger for the urban Harappan collapse was probably the decline of the summer monsoon, the agricultural Harappan economy showed a large degree of adaptation to water 1173

1174 availability. The long-lived survival of Late Harappan cultures until ca. 3,200 years ago under a

1175 drier climate and less active fluvial network is the subject of the present study and further

1176 ongoing efforts (e.g., Kotlia et al., 2017; Petrie et al., 2017) that seek to understand the

1177 variability in hydroclimate and moisture sources across the Indus domain and how these relate to

agricultural adaptations.

|  <li>Figure Captions</li> <li>Fig. 1. Physiography, winds and precipitation sources for the Harappan domain. The dominant source during summer monsoon is the Bay of Bengal while Western Disturbances provide the moisture during winter. The extent of the Indus basin and Ghaggar-Hakra (G-H) interfluve are shown with purple and brown masks, respectively. Locations for the cores discussed in the text are shown.</li> <li>Fig. 2. Geographical regions and rivers of the Indus domain discussed in text.</li> <li>Fig. 3. Modern seasonal climatology for South Asia. Average precipitation as well as wind direction and intensity for the summer (June-July-August or JJA) and winter (December-January-February or DJF) months are presented in the left and right panels, respectively. Note the differences in scales between panels for both rainfall and winds. Data used come from the</li>  |                                               |
|------------------------------------------------------------------------------------------------------------------------------------------------------------------------------------------------------------------------------------------------------------------------------------------------------------------------------------------------------------------------------------------------------------------------------------------------------------------------------------------------------------------------------------------------------------------------------------------------------------------------------------------------------------------------------------------------------------------------------------------------------------------------------------------------------------------------------------------------------------------------------------------------------------------------|-----------------------------------------------|
| 1195 ERA-40 reanalysis dataset (Uppala et al., 2005) for winds (averaged from 1958-2001) and the                                                                                                                                                                                                                                                                                                                                                                                                                                                                                                                                                                                                                                                                                                                                                                                                                       |                                               |
| 1196 TRMM dataset (Huffman et al., 2007) for rainfall (averaged from 1998-2014). The white box
1197 encompasses the upper G-H interfluye.                                                                                                                                                                                                                                                                                                                                                                                                                                                                                                                                                                                                                                                                                                                                                                           | s                                      |
| 1198                                                                                                                                                                                                                                                                                                                                                                                                                                                                                                                                                                                                                                                                                                                                                                                                                                                                                                                   |                                               |
| 1199Fig. 4. Holocene variability in plankton communities as reflected by their sedimentary DNA                                                                                                                                                                                                                                                                                                                                                                                                                                                                                                                                                                                                                                                                                                                                                                                                                         | Liviu Giosan 10/17/2018 1:41 PM               |
|  <li>factor loadings (panels marked a through c) and winter mixing-sensitive % G. falconensis (panel</li> <li>marked d) in core Indus 11C in the NE Arabian Sea. Relative chlorophyll biosynthesis proteins</li> <li>abundances are also shown. Sea level points are from Camoin et al. (2004); SSTs are from</li> <li>Doose-Rolinski et al. (2001); and G. falconensis census from the NW Arabian Sea is from</li> <li>Schulz et al. (2002). Triangles show radiocarbon dates for core Indus 11C. The period</li> <li>corresponding to the Early Neoglacial Anomalies (ENA) is shaded in red hues.</li>                                                                                                                                                                                                                                                                                        | Deleted: 3                                    |
| 1207 Fig 5 Northern Hemisphere hydroclimatic conditions since the middle Holocene. The period                                                                                                                                                                                                                                                                                                                                                                                                                                                                                                                                                                                                                                                                                                                                                                                                                          |                                               |
|  <li>corresponding to the Early Neoglacial Anomalies (ENA) interval is shaded in red hues. From</li> <li>high to low (panels marked a trough i): (a) Greenland dust from non-sea-salt K+ showing the</li> <li>strength of the Siberian Anticyclone (O'Brien et al., 1995); (b) NAO proxy reconstruction (Olser</li> <li>et al., 2012) and (c) negative NAO-indicative floods in S Alps (Wirth et al., 2013); (d) grainsize-</li> <li>based hurricane reconstruction in the N Atlantic (van Hengstum et al., 2016); (e)</li> <li>interhemispheric temperature anomaly (Marcott et al., 2013); (f) ITCZ reconstruction at the</li>                                                                                                                                                                                                                                                                   | Liviu Giosan 10/17/2018 1:41 PM
| 1214 Cariaco Basin (Haug et al., 2011); (g) winter monsoon ancient DNA-based reconstruction for the
1215 NE Arabian Sea (this study – in purple); (h) speleothem $\delta^{18}$ O-based precipitation reconstruction
1216 for northern Levant (Cheng et al., 2015); and (i) stacked lake isotope records as a proxy
1217 precipitation-evaporation regimes over Middle East and Iran (Roberts et al., 2011).
1218                                                                                                                                                                                                                                                                                                                                                                                                                                                                                           | Deleted: Marcot                               |
| 1219 Fig. 6 . Monsoon hydroclimate changes since the middle Holocene and changes in settlement                                                                                                                                                                                                                                                                                                                                                                                                                                                                                                                                                                                                                                                                                                                                                                                                                  |                                               |
|  <li>distribution on the Ghaggar-Hakra interfluve. From high to low (panels marked a trough f): (a)</li> <li>variability in summer monsoon calculated as 200-year window moving standard deviation of the</li>                                                                                                                                                                                                                                                                                                                                                                                                                                                                                                                                                                                                                                                                                                | Liviu Giosan 10/17/2018 1:41 PM

- 1226 detrended monsoon record of Katahayat et al. (2017) and (b) the speleothem  $\delta^{18}$ O-based summer
- 1227 monsoon reconstruction of Katahayat et al. (2017); (c) lacustrine gastropod  $\delta^{18}$ O-based summer
- 1228 monsoon reconstruction (Dixit et al., 2014); (d and e) changes in the number of settlements on
- 1229 the Ghaggar-Hakra interfluve as a function of size and location; and (f) winter monsoon ancient
- 1230 DNA-based reconstruction for the NE Arabian Sea (this study in purple). The period
- 1231 corresponding to the Early Neoglacial Anomalies (ENA) is shaded in red hues and durations for
- 1232 Early (E), Mature (M) and Late (L) Harappan phases are shown with dashed lines.